# Orientation dependent molecular electrostatics drives efficient charge generation in homojunction organic solar cells

Yifan Dong[1,11], Vasileios C. Nikolis[2,10,11], Felix Talnack[3], Yi-Chun Chin [4], Johannes Benduhn [2], Giacomo Londi [5], Jonas Kublitski [2], Xijia Zheng[1], Stefan C. B. Mannsfeld [3], Donato Spoltore[2], Luca Muccioli[6], Jing Li[7], Xavier Blase[7], David Beljonne[5], Ji-Seon Kim [4✉], Artem A. Bakulin [1], Gabriele D'Avino[7✉], James R. Durrant [1,8✉] & Koen Vandewal [9✉]

Organic solar cells usually utilise a heterojunction between electron-donating (D) and electron-accepting (A) materials to split excitons into charges. However, the use of D-A blends intrinsically limits the photovoltage and introduces morphological instability. Here, we demonstrate that polycrystalline films of chemically identical molecules offer a promising alternative and show that photoexcitation of α-sexithiophene (α-6T) films results in efficient charge generation. This leads to α-6T based homojunction organic solar cells with an external quantum efficiency reaching up to 44% and an open-circuit voltage of 1.61 V. Morphological, photoemission, and modelling studies show that boundaries between α-6T crystalline domains with different orientations generate an electrostatic landscape with an interfacial energy offset of 0.4 eV, which promotes the formation of hybridised exciton/charge-transfer states at the interface, dissociating efficiently into free charges. Our findings open new avenues for organic solar cell design where material energetics are tuned through molecular electrostatic engineering and mesoscale structural control.

[1] Department of Chemistry and Centre for Processable Electronics, Imperial College London, London W12 0BZ, UK. [2] Dresden Integrated Centre for Applied Physics and Photonic Materials (IAPP) and Institute for Applied Physics, Technische Universität Dresden, Nöthnitzer Str. 61, 01187 Dresden, Germany. [3] Center for Advancing Electronics Dresden (cfaed) and Faculty of Electrical and Computer Engineering, Technische Universität Dresden, Helmholtzstr. 18, 01069 Dresden, Germany. [4] Department of Physics and Centre for Processable Electronics, Imperial College London, London SW7 2AZ, UK. [5] Laboratory for Chemistry of Novel Materials, University of Mons, Place du Parc 20, 7000 Mons, Belgium. [6] Department of Industrial Chemistry, University of Bologna, Viale Risorgimento 4, 40136 Bologna, Italy. [7] Université Grenoble Alpes, CNRS, Grenoble INP, Institut Néel, 25 Rue des Martyrs, 38042 Grenoble, France. [8] SPECIFIC, College of Engineering, Swansea University, Bay Campus, Swansea SA1 8EN, UK. [9] Institute for Materials Research (IMO-IMOMEC), Hasselt University, Wetenschapspark 1, 3590 Diepenbeek, Belgium. [10] Present address: Heliatek GmbH, Treidlerstraße 3, 01139 Dresden, Germany. [11] These authors contributed equally: Yifan Dong, Vasileios C. Nikolis. ✉email: ji-seon.kim@imperial.ac.uk; gabriele.davino@neel.cnrs.fr; j.durrant@imperial.ac.uk; koen.vandewal@uhasselt.be

In contrast to most inorganic semiconductors, the low dielectric constants of organic conjugated molecules lead to the formation of tightly bound electron–hole pairs, namely excitons, upon illumination. As a result, early organic solar cells (OSCs) employing only one absorber material could hardly reach a power conversion efficiency (PCE) of 0.1%[1]. Efficient dissociation of excitons could only be achieved by combining two organic semiconductors to form a heterojunction, with one acting as an electron donor (D) and the other one as an electron acceptor (A)[2]. An energy level offset between D and A leads to the formation of intermolecular charge-transfer (CT) states, an electronic state optically coupled to the ground state, in which the electron resides on A and the hole on D, playing a key role in free charge carrier generation[3,4]. Based on the D–A concept, the PCE for D–A single junction OSCs has increased from 0.1% to over 18%[5]. This efficiency improvement has resulted from multiple factors, ranging from material synthetic design, dielectric constant tuning, interface modification, morphological engineering etc[6–9]. However, a downside of this approach is that the D–A energy offset also induces significant voltage losses, which limit the highest achievable open-circuit voltage ($V_{OC}$)[10]. Moreover, optimisation of the blend morphology, finding the best balance between exciton diffusion length and efficient charge transport, as well as preserving this morphology over time, has always been challenging[11,12]. Therefore, fabricating single material devices, as homojunction OSCs (HOSCs), remains as a highly attractive alternative, which offer the potential for relatively easy processing and morphology control, and could also lead to high photovoltages.

Rare examples of such devices mostly rely on combining D and A moieties inside a single chemical entity as a large molecule or a copolymer with a D–A architecture[13,14]. However, the complexity in synthesis and processing, as well as ultrafast recombination losses within such structures, has to date limited their further development[15]. Here, we instead study a thiophene-based small molecule, α-sexithiophene (α-6T), which is widely known only as an electron donor and a p-type organic semiconductor, i.e., it is devoid of the D–A dual nature[16,17]. It was previously shown by Duhm et al. that controlling the orientation of α-6T can lead to different interface energetics as a result of long-range intermolecular electrostatic interactions[18,19]. However, the tantalising idea of using molecular electrostatics engineering to fabricate efficient photovoltaic devices has not been realised in practice until now.

In this work, we report the efficient photocurrent generation in α-6T-based HOSC, in the absence of an electron accepting material, reaching an external quantum efficiency (EQE) of 44% and a $V_{OC}$ of 1.61 V. Transient absorption (TA) spectroscopy measurements show that the photoexcitation of pristine α-6T

films leads to efficient and ultrafast charge carrier generation from excitons, indicating that charge generation happens in the bulk of the pristine α-6T. Using specific processing parameters, we fabricated films with two molecular orientations, mainly standing and mainly lying, and measured an offset of 0.4 eV between the ionisation potentials at these two orientations. State-of-the-art calculations based on embedded many-body theories revealed that this energy offset dictated by intermolecular electrostatic interactions persists at standing/lying grain boundaries, where it promotes the formation of low-lying excitations with hybrid intramolecular/CT character. The population and dissociation of these interfacial states provide a rationale for the high charge generation quantum yield in α-6T-based HOSCs.

## Results

**Device performance.** Being one of the archetypal organic semiconductors, α-6T has been employed as an electron-donating material and paired with various fullerene and non-fullerene acceptors in OSCs[16,17]. Here, we test the charge generation efficiency in α-6T by fabricating HOSCs, where α-6T itself is expected to facilitate photon absorption and charge generation. The device was fabricated at room temperature by evaporating an α-6T layer between the bottom (indium tin oxide, ITO) and the top contact (silver, Ag). BPhen was also inserted as the cathode buffer layer (BL), alone or combined with other BL materials (Rubrene, C545T, DBzA, TCTA, TPBA, TBPe, TPBI—chemical names are given in Supplementary Table 1) that are commonly used in OSCs and organic light emitting diodes (Fig. 1a)[20,21]. The selection of the BLs adjacent to α-6T was based on their lowest unoccupied molecular orbital (LUMO) energy, being comparable to that of α-6T (Supplementary Table 1), and their use focuses on the improvement of contact selectivity and device performance[20,22]. In the case that the LUMO of the BL material (TCTA, for instance) is much higher than that of α-6T, we expect tunnelling to aid electron transport, considering the low thickness of the BL layer (10 nm), as well as the reported high roughness of the α-6T layer[16], which can lead to a discontinuous/inhomogeneous thick BL layer deposited on of it. Figure 1b, c summarise the EQE spectra and the current–voltage characteristic curves of the investigated devices. The measured EQEs are between 35% and 50% for all used BL materials, being 40–45% for the majority of them (Fig. 1b).

While no obvious D–A interfaces are present in the devices, there is the possibility that charge generation takes place at the α-6T/BL interface. However, the fact that EQE values do not strongly depend on the specific nature of the BL, suggest that charge generation occurs in the bulk of the α-6T film. As will be shown below using TA spectroscopy, this is indeed the case.

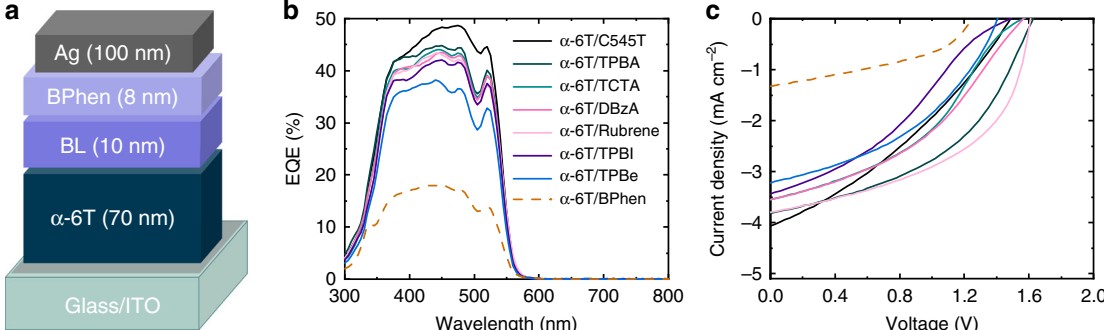

**Fig. 1 Organic solar cells based on α-6T and various buffer layers. a** Device architecture of the investigated devices. BPhen and an additional buffer layer (BL) are used between α-6T and the top contact (Ag). The numbers in the parentheses denote the layer thickness in nanometre. **b** External quantum efficiency (EQE) spectra and **c** current–voltage characteristic curves of solar cells employing α-6T and various BL materials.

**Table 1 Photovoltaic parameters of organic solar cells based on α-6T with various buffer layers.**

| Device structure | $V_{OC}$ (V) | $J_{SC}$ (mA cm$^{-2}$) | FF (%) | PCE (%) |
|---|---|---|---|---|
| ITO/α-6T/BPhen/Ag | 1.25 | 1.4 | 41.4 | 0.7 |
| ITO/α-6T/Rubrene/BPhen/Ag | 1.61 | 3.6 | 50.2 | 2.9 |
| ITO/α-6T/C545T/BPhen/Ag | 1.46 | 3.8 | 34.7 | 1.9 |
| ITO/α-6T/DBzA/BPhen/Ag | 1.57 | 3.3 | 39.9 | 2.1 |
| ITO/α-6T/TCTA/BPhen/Ag | 1.57 | 3.3 | 39.4 | 2.1 |
| ITO/α-6T/TPBA/BPhen/Ag | 1.61 | 3.6 | 47.0 | 2.8 |
| ITO/α-6T/TBPe/BPhen/Ag | 1.41 | 3.0 | 42.5 | 1.8 |
| ITO/α-6T/TPBI/BPhen/Ag | 1.50 | 3.2 | 33.3 | 1.6 |

However, it should be noted that the use of extra BLs improves significantly the $V_{OC}$ (up to 1.61 V) compared to the optical gap of α-6T (2.33 eV, Supplementary Fig. 1), mainly due to a reduction of the non-radiative recombination in those devices, leading to total energy losses of 0.72 eV in the device with Rubrene and TPBA (Supplementary Table 2). Except for the device with only BPhen, the sensitively measured EQE spectra of all other devices do not show any subgap absorption features, implying the absence of low-energy CT states originating from the α-6T/BL interface, which could drive charge generation and recombination (Supplementary Fig. 2). Overall, the device with the best performance is the one employing rubrene as BL, exhibiting the highest $V_{OC}$ (1.61 V), FF (0.52), and a $J_{SC}$ of 3.6 mA cm$^{-2}$, resulting in a PCE of 2.9% and an EQE of 44%, impressive for a device acting as a HOSC (Fig. 1b and Table 1).

**Charge generation in α-6T-based devices and films**. We turn now to consideration of the mechanism of charge generation in these devices. While the spectroscopic study of photoexcitation of α-6T thin films dates back to the 1990s[23], the photophysical mechanism of charge generation in the α-6T-based HOSC is still unclear. We first observed that the photoluminescence (PL) of α-6T/rubrene bilayers was the sum of the PL of the individual layers (Supplementary Fig. 4), indicating negligible exciton quenching at the α-6T/rubrene interface. This observation strengthens the hypothesis, suggested by the device study above, that charge generation may take place inside the α-6T layer. To investigate this further, we employed low excitation density TA spectroscopy to investigate α-6T devices and thin films on ITO. Figure 2a shows the TA spectra of an α-6T film fabricated with the same processing conditions as the α-6T-based HOSC with rubrene as BL. In these spectra three negative bands peaking at 525, 593 and 650 nm are apparent, as well as two sharp positive bands at 508 and 780 nm. In previous studies, the features at 525 and 508 nm have been assigned to electroabsorption (EA) signals associated with CT states[24]. We independently measured the EA of the α-6T HOSC (Supplementary Fig. 5) and validated the resemblance between the EA and TA spectra, which further suggests that the strong signal at 525 and 508 nm in TA arises from the initially generated excitons having a high-degree of charge transfer character, most likely associated with the hybridisation between exciton and CT states (see further discussions below). The observation of similar EA signals has been reported for D–A OSCs[25,26]. The 780 nm positive absorption feature has previously been assigned to α-6T charges by Watanabe et al. and others (see also discussion below)[27–30]. Comparison with the absorption and emission spectra (Fig. 3a, b) indicate that ground state bleaching is also likely to contribute to the 525 nm feature, whilst the negative features around 593 and 650 nm can be assigned to stimulated emission from α-6T singlet excitons.

It is apparent from Fig. 2a that the time evolution of the TA signal implies a decay of the negative exciton features at 593 and 650 nm, together with a partial decay of the 508 nm EA feature

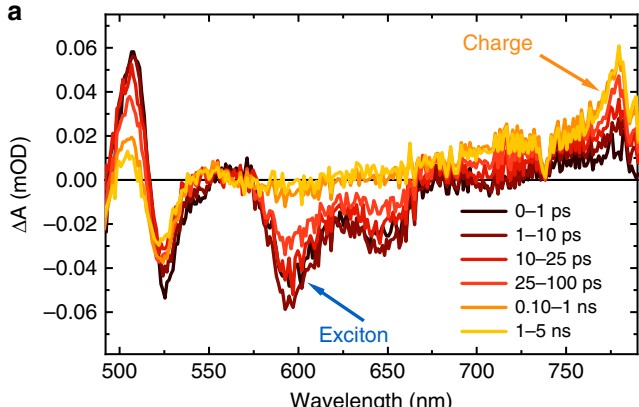

**a**

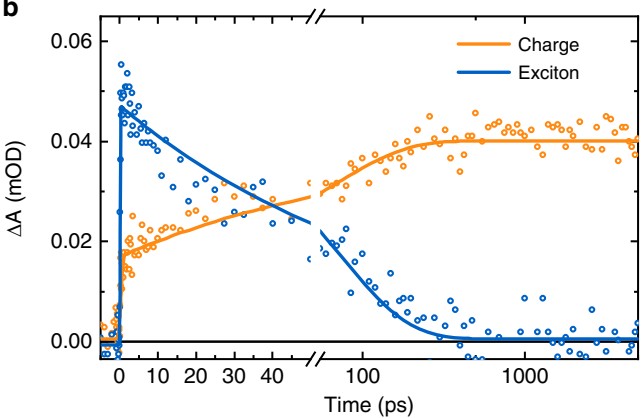

**b**

**Fig. 2 Transient absorption (TA) characterisation for the pristine α-6T thin film. a** TA spectra for pristine α-6T thin film employing a pump wavelength of 450 nm; **b** TA kinetics at 593 nm (blue) and 780 nm (orange) representing stimulated emission (SE) and photoinduced absorption (PIA) signals, respectively, where the lifetimes of exciton decay and charge generation can be extracted from individual dynamics. The solid lines are exponential fitting for the raw data (dots). Low (5 μJ cm$^{-2}$) excitation fluences were used to minimise exciton–exciton annihilation and bimolecular recombination processes (see Supplementary Figs. 6 and 8 for details).

and a growth of the 780 nm signal attributed to charge carrier absorption. Fitting the kinetics at 593 nm with a mono-exponential decay gives a lifetime of 71 ps, assigned to the exciton decay time (Fig. 2b, blue circles). The kinetics at 780 nm exhibits a prompt growth within our instrument response (~200 fs), followed by a slower growth, with fits to this slower phase yielding a mono-exponential rise time of 69 ps (Fig. 2b, orange circles). The agreement between the exciton decay lifetime (measured at 593 nm) and charge absorption growth time (measured at 780 nm) indicates that charge generation correlates with the exciton decay. The observation of EA absorption features

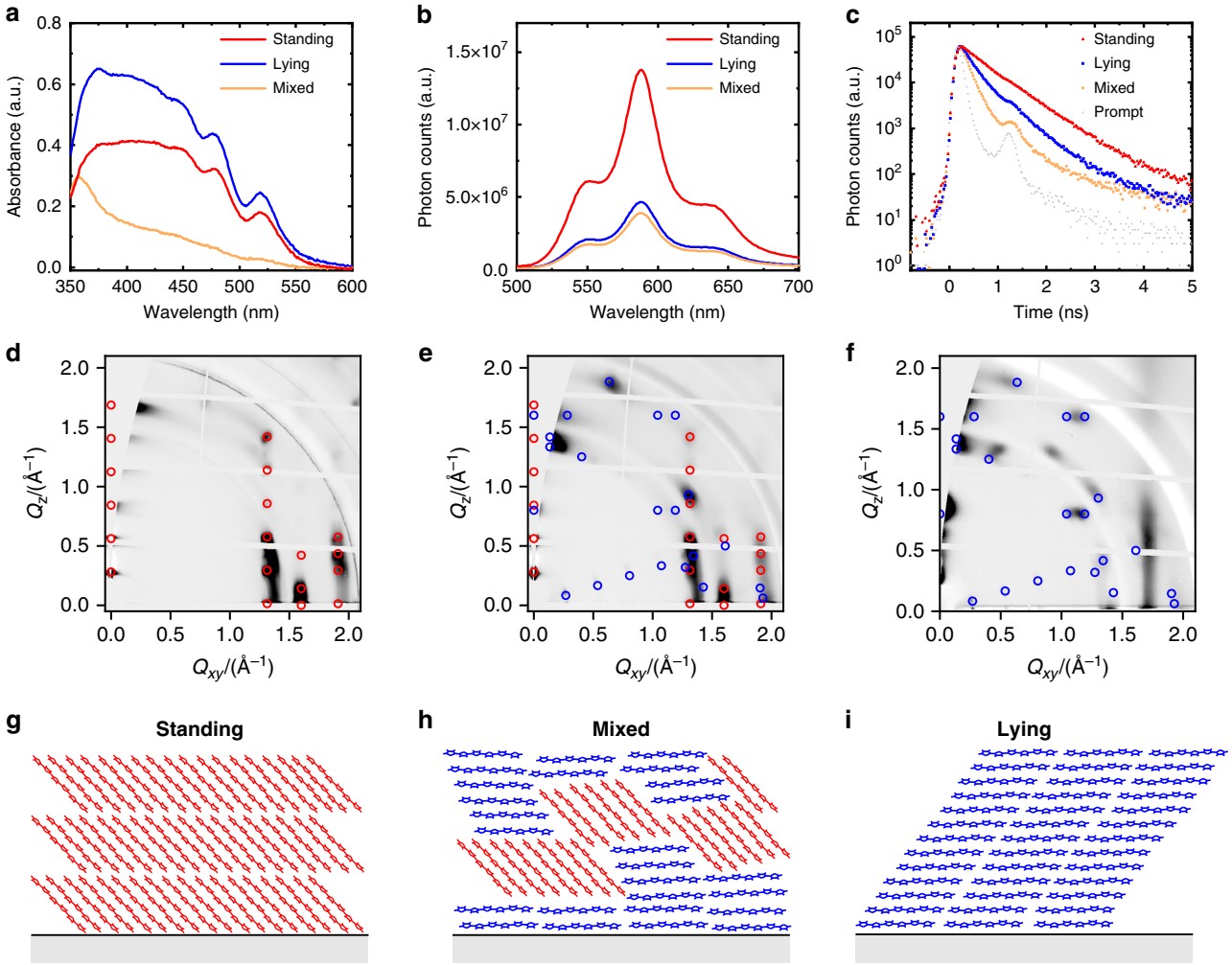

**Fig. 3 Morphological and spectroscopic characterisation for the orientation of α-6T thin films. a** Absorbance spectra for α-6T thin films with different molecular orientations where the lying molecules show the highest absorbance and the mixed orientation film lies in between the lying and standing samples. **b** Photoluminescence (PL) spectra normalised with the absorbance at the excitation wavelength of 450 nm for α-6T thin films with different molecular orientations where the PL quenching is observed in the mixed molecules. **c** Time-correlated single photon counting for α-6T thin films with different molecular orientations (standing, mixed and lying) revealing a faster PL decay in the presence of both standing and lying orientations. The grey dots indicate the prompt decay (instrument response function), which in the case of the mixed and lying orientations induces an artefact at 1.3 ns. **d–f** Grazing-incidence wide-angle X-ray scattering (GIWAXS) diffraction images showing that the molecular orientation of α-6T thin films can be tuned with specific processing conditions. The peaks indicated by red and blue circles originate from crystallites with standing and lying molecules, respectively. **g–i**, schematic morphology for α-6T thin films with different molecular orientations (standing, mixed, and lying).

at the early time suggests that the initially generated α-6T excitons exhibit at least partial CT state character, as discussed further below. The decay of this EA feature on the timescale of free carrier generation is most likely associated with field screening as charges localise in separate domains, discussed further below[25]. We further carried out TA measurements for pump fluences from 5–20 μJ cm$^{-2}$ (Supplementary Fig. 6a). The kinetics at 780 nm exhibit a strong dependence on the pump fluence, with decay dynamics accelerating at higher fluences characteristic of bimolecular recombination. Fitting the kinetics gives a bimolecular rate constant on the order of $10^{-11}$ cm$^3$ s$^{-1}$ (Supplementary Fig. 6b), which is of similar magnitude or slower than that observed in typical bulk heterojunction OSCs[31–33]. This fluence dependence therefore suggests that this PIA at 780 nm originates from free charge carriers rather than being caused by the formation of CT or triplet states[30,34–36]. At low pump fluences, a negligible decay of this feature is observed for time delays up to 6 ns (plateau in Fig. 2b), indicative of long-lived free charge carrier generation.

Turning to the α-6T-based HOSC with rubrene as BL, similar exciton decay lifetimes were observed (Supplementary Figs. 7 and 8). Except for a slightly faster charge recombination, all the spectral features are nearly identical with those in pristine α-6T thin films on ITO, indicating that efficient free charge carrier generation is not only present in devices but also in α-6T thin films. This confirms our conclusions above that device interfaces are not the origin of charge generation and indicates rather that this is a bulk phenomenon inherent to α-6T, one that is not usually observed in pristine organic materials.

Efficient free charge generation in pristine organic semiconductors without intermolecular CT character has been rarely reported to date (unless very high photon energies are used[37]) due to the strong binding energy of excitons in such materials (typically estimated as several hundreds of meV). Only a few spectroscopic studies have suggested ultrafast photogeneration of free charge carriers in pristine organic small molecule films, with these reporting only low EQEs of less than 10% and more often less than 1%[37–40]. For example, Keiderling et al. reported charge

generation in pristine PCBM thin films. This however results in rather low efficiency photocurrent generation in PCBM HOSCs, as reported by Burkhard et al.[41,42]. Other organic materials reported to exhibit charge or photocurrent generation include MEH-PPV, $C_{60}$ and ZnPc[38,43,44]. However, in those reports either the device performance was poor or the mechanisms for the fairly limited intrinsic charge generation have remained unexplained. In the next sections, we carry out further measurements to understand what underlines the efficient charge photogeneration in pristine α-6T films.

**Effect of orientation on energetics**. It is known that charge carrier energetics in organic semiconductor devices is strongly intertwined with the molecular organisation in the solid state[19]. For instance, Duhm et al. have reported a 0.4 eV difference in the ionisation potential of α-6T films as the orientation transitions from standing to lying[18,45]. Indeed, the presence of a possible energy offset between domains characterised by different molecular orientations may determine the favourable conditions for the charge separation. It is therefore essential to unravel the relationship between electronic properties, molecular organisation as well as the device processing conditions that confer the possibility to control the first two aspects, as we address below.

A fast deposition of α-6T (1 Å/s) on unheated substrate, as employed for the films and HOSCs reported in Figs. 1 and 2 above, results in a mixture of different orientations (standing and lying) with respect to the ITO substrate[46]. In addition, we also prepared films with mainly standing molecules, as well as films containing mainly lying molecules, by employing fabrication parameters which have been used in previous reports (see also 'Methods')[46–48]. In the lying orientation, the absorbance is maximised due to the parallel alignment between the transition dipole moment of the molecule and the electric field vector of incident light (Fig. 3a)[49,50]. Since the amount of lying molecules is lower in the mixed and standing films, the absorbance decreases accordingly (Fig. 3a). The mixed orientation lies in between the two, indicating the co-existence of both orientations (see GIWAXS data in Fig. 3d–f, discussed in detail below). While the absorption spectra vary in shape and intensity for differently oriented films, the PL spectra retain the same shape since they originate from the lowest excited state after the excitations undergoing a significant vibrational relaxation. However, PL and time-correlated single photon counting (TCSPC) show a quenching and a faster PL decay in the mixed film in comparison with the other two orientations (Fig. 3b, c), implying that an enhanced exciton quenching mechanism is present in the mixed film containing both standing and lying α-6T phases.

Grazing-incidence wide-angle X-ray scattering (GIWAXS) was employed to investigate the morphology of the films with standing, mixed and lying molecular orientations, as shown in Fig. 3d–f. For the films prepared at high temperatures (Fig. 3d), the lamellar stacking peaks are clearly visible in the out-of-plane direction, corresponding to standing molecules. In addition, the (H 1 1), (H 2 0) and (H 2 1) Bragg rods visible at 1.31, 1.60, and 1.91 Å$^{-1}$ are typical for the herringbone motif adapted by the α-6T in the low temperature polymorph. The positions of the red circles were derived from the unit cell reported for the low-temperature polymorph by Horowitz et al.[51]. The same naming for the unit cell parameters was used herein. In Fig. 3f, α-6T deposited on top of an ITO substrate with a thin film of copper iodide (CuI, 2 nm) are shown. Compared to Fig. 3d, the lamellar peaks shifted to the in-plane direction, corresponding to lying molecules. The rest of the peaks indicate that the unit cell lies down in two ways. One in which the b-axis of the unit cell is perpendicular to the surface and one in which both short axes of

the unit cell have an out-of-plane contribution, with the (3 1–1) plane being roughly parallel to the substrate. The diffraction image for mixed films deposited at room temperature with a high evaporation rate, shown in Fig. 3e, is a superposition of the diffraction images of the films with lying and standing orientation, shown in Fig. 3d and f, respectively. This is clearly visible by the existence of the lamellar peaks in the out-of-plane direction (red circles, standing orientation) and the in-plane direction (blue circles, lying orientation). In addition, to the Bragg-rods visible in-plane ($q_{xy} = 1.31$, 1.60 and 1.91 Å$^{-1}$), which originate from the standing orientation, peaks can be seen at positions corresponding to lying orientations, further showing that both molecular orientations are present in these films. Analysis of the width of the GIWAXS peaks (Supplementary Fig. 11) provides a lower limit of the average size of standing and lying crystallites, which for the mixed film are roughly 10 nm, i.e., they are comparable with the exciton diffusion length of α-6T[52].

We then carried out energy level measurements for thin films of standing or lying orientation on ITO using ambient photoemission spectroscopy (APS), as shown in Fig. 4a. A highest occupied molecular orbital (HOMO) level of 4.8 eV was observed for the standing film, compared to a value of 5.2 eV for the lying film. The APS measurements confirm the existence of an energy offset δ ~ 0.4 eV between the HOMO levels of samples of lying and standing α-6T molecules.

APS measurements determine the energetics of charge carriers at the surface of the two films of either standing or lying molecules. However, the energy landscape at the boundary between two domains may well be affected by the different nature of the environment (bulk-like, without interface to air). To complement APS measurements, we performed embedded many-body GW calculations, which was recently proved able to reach quantitative accuracy in the determination of energy levels in molecular solids[53]. We have explicitly considered a model interface between lying and standing molecules, obtaining an inter-domain energy offset (δ ~ 0.4 eV) that is in close agreement with the one determined by APS for the two films with different molecular orientations. Our calculations showed that this energy offset is determined by the different orientations of the molecular quadrupoles in each side of the grain boundary, which dictate a step in the electrostatic potential across the interface (Fig. 4b, c) that affects occupied and unoccupied levels[54]. As sketched in Fig. 4d, standing and lying α-6T molecules behave therefore as the electron donating and the accepting components of a conventional organic heterojunction, consistent with the efficient charge generation observed in our TA data above. In a working device, hole and electron transport to electrodes would then take place in the standing and lying domains, respectively, exploiting the intrinsic ambipolar character of α-6T films[55,56] and consistent with the reasonably slow bimolecular recombination observed for this system.

The presence of a 0.4 eV energy offset between standing and lying domains may result in the occurrence of CT transitions, where an electron is transferred from the standing to the lying domain, in the lowest-energy tail of the absorption spectrum. These states could act as a gateway for efficient charge separation, as in conventional organic heterojunctions. To gain insight into low-energy optical excitations, we performed state-of-the-art embedded Bethe–Salpeter equation calculations, which allowed us to accurately describe both Frenkel and CT excitations in large molecular systems[57]. The calculated absorption spectrum in Fig. 5 reveals the complexity of the excited-state manifold, which is heavily affected by delocalisation and hybridisation between molecular (Frenkel, with high oscillator strength) and CT excitons. The brightest exciton is computed at 2.42 eV (hybrid Frenkel-CT, see Fig. 5c), in excellent agreement with the

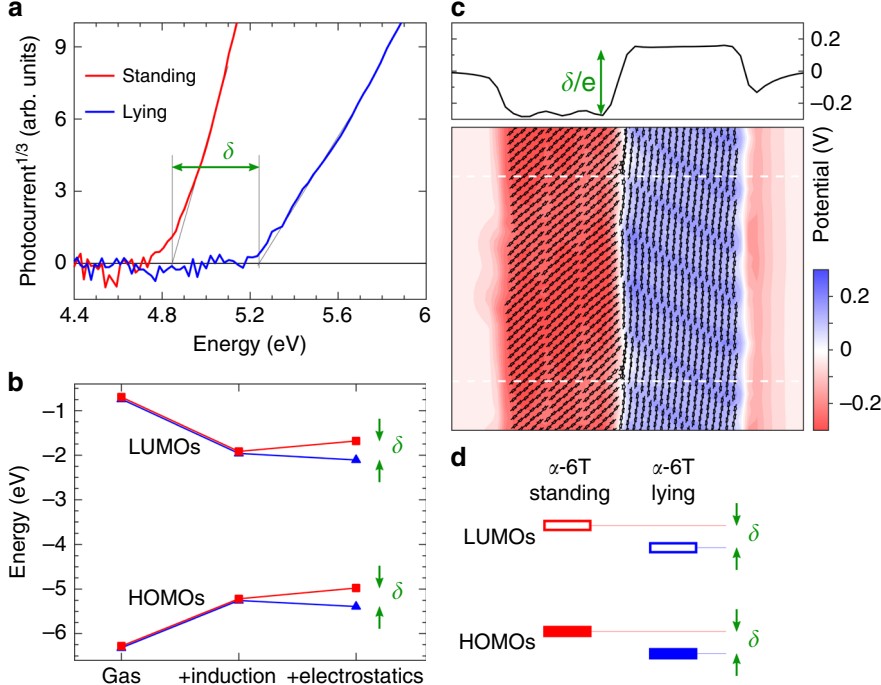

**Fig. 4 Charge transport energy levels in α-6T thin films. a** Ambient photoemission spectroscopy (APS) results for α-6T thin films with standing and lying orientations presenting an energy difference δ ∼ 0.4 eV between their HOMO levels (referenced to the vacuum level). **b** Evolution of HOMO and LUMO levels calculated from embedded *GW* calculations for an interface between two domains with standing and lying α-6T molecules. Results are presented by progressively adding induction (dielectric response) and electrostatic intermolecular interactions to gas-phase levels. This shows that the offset δ between standing and lying molecules is entirely sourced by electrostatics. **c** Maps of the electrostatic potential illustrating the step-like variation across the standing-lying interface. **d** Sketch of the energy levels of standing and lying α-6T molecules, playing the role of electron donor and acceptor component, respectively.

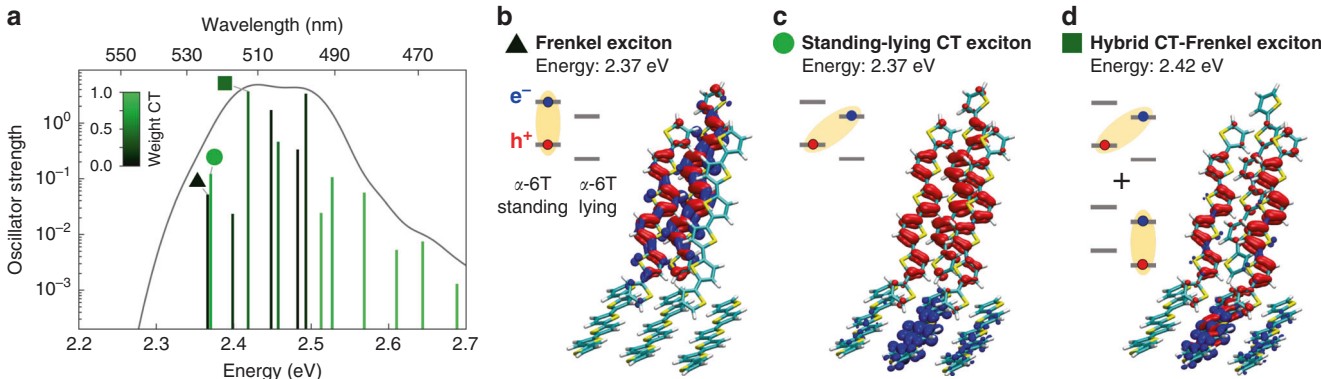

**Fig. 5 Optical excitations from embedded Bethe–Salpeter calculations. a** Absorption spectrum of the standing–lying α-6T interface. Bar colours quantify the weight of inter-layer CT states of each excitation, showing that the lowest-energy region of the spectrum presents states that have a pronounced charge separation, with hole and electron localised in the standing and lying domains, respectively. The energy of these states with spatially separated charges is determined by the interfacial energy offset and can act as a gateway for an efficient charge splitting. **b–d** Electron–hole density plots and the corresponding simplified energy-level sketches of representative low-energy excitations.

experimental absorption spectrum (Fig. 3g). Our calculations further reveal that one of the two nearly degenerate lowest-energy excitations, placed just ~50 meV below the brightest exciton, is an inter-domain CT state with the electron in the lying domain and the hole in the standing domain of the interface (see Fig. 5b). It is likely that thermal molecular motion, not included in our modelling, may reshuffle the nature of these lowest-energy excitations, leading to a dynamic interconversion between Frenkel and CT states, the latter providing the initial step for an efficient charge separation.

## Discussions
Our combined experimental and theoretical analysis draws a close analogy between the homojunction between crystalline domains with standing–lying orientations and D–A heterojunctions in conventional OSCs, even though in our single component system the interfacial energy offset is solely determined by intermolecular electrostatic interactions, and not by any chemical difference between the two species as in heterojunctions. Such an energy offset of electrostatic origin between the standing and lying orientations determines high-lying CT states that are nearly

degenerate with bright excitons and that are, therefore, hardly resolved experimentally. Indeed, no extra low-energy CT absorption band is observed in the sensitively measured EQE spectra (Supplementary Fig. 2). Furthermore, this agrees with our TA data for the α-6T film (Fig. 2) where an indication for the fast formation of exciton-CT hybrid states is found in the charge-induced EA signal at 525 nm and is crucial towards charge generation. This type of high-lying hybrid exciton/CT states have also recently been proposed to be responsible for the low voltage losses in the state-of-the-art D–A OSCs[58]. It has also recently been suggested that CT separation may be assisted by the electrostatic interfacial fields in low energy offset, high performing OSCs based on a conventional D–A heterojunction[59]. The population of the inter-domain CT excitations following light absorption appears to be the key for the unprecedented charge generation efficiency not only for a single component molecular film and devices, but also for D–A heterojunction OSCs.

We have investigated the origin of the remarkably high device performance in α-6T-based HOSCs from various aspects including TA and APS as well as simulations. Using a single material presents the advantage of achieving high voltage and allows great potential for controlled and reproducible morphology and performance. We report a record high EQE of up to 40% and a $V_{OC}$ of 1.61 V for an HOSC employing only the donor organic molecule α-6T composed of identical crystallites with a distribution of relative orientations.

We have found that charge separation is driven by the electrostatic landscape generated at the interface between lying and standing α-6T molecules, which creates an energy offset of about 0.4 eV between these two domains. We attribute this to be the reason behind charge separation in pristine α-6T films: the standing domain plays the role of the donor and the lying domain is the acceptor. A generalisation of this result is that the ordered morphology and the contact between different crystal facets are crucial to dissociate excitons, and that grain boundaries in single component polycrystalline systems could be exploited as 'heterojunctions'. Engineering morphology in the preferred way is the key to generate energy offsets in single materials and facilitate charge generation pathway.

This work reshapes the common understanding in the role of 'donor' and 'acceptor' in OSCs where this character is not only fixed by the primary chemical structure of the molecules but can also be modulated by interfacial packing and electrostatics. This further widens the broad potentiality in donor and acceptor materials in OSCs.

## Methods

**Thin films and devices fabrication**. *Thin film samples fabrication*: The thin film samples were prepared on glass substrates with a sputtered ITO layer. Thin films of either 20 or 60 nm were fabricated for each orientation. Both thicknesses were used for the APS measurements, while GIWAXS and TAS were performed on samples, which had a thickness of 60 nm. Herein, three different processing methods were used, leading to three different morphologies. When α-6T was deposited at low evaporation rates (~0.1 Å/s) on substrates heated at 100 °C, the α-6T molecules orientate with their long axis almost perpendicular to the substrate, adopting an upright 'standing' orientation. If a thin (2 nm) layer of CuI was used as templating layer on top of the ITO electrode, the α-6T molecules deposited on CuI tend to orientate in parallel to the substrate, leading to a 'lying' orientation. This is due to the stronger interaction between the π-conjugated system of α-6T and the *d* orbitals of CuI, which lead to the templating effect[48]. Finally, depositing α-6T at room temperature and high evaporation rate (~1 Å/s) resulted in 'mixed' films where the edge-on and face-on crystal phases are coexistent.

*Device fabrication*: The solar cells shown in this publication were processed by thermal evaporation in a custom-made vacuum system (Kurt J. Lesker, USA) with a base pressure of $10^{-7}$ mbar. During a processing run, different masks and movable shutters enable the variation of the device stacks or processing parameters, offering the possibility to produce and compare various devices at the same processing conditions. Each device was fabricated onto either clean glass substrates or substrates with pre-structured ITO (Thin Film Devices, USA), which underwent an ozone treatment for cleaning before being transferred into the vacuum chamber.

Every investigated device was bottom illuminated, employing ITO as the anode and a 100-nm-thick Ag as the cathode. The full device structure of the investigated devices is as follows: ITO/α-6T(70 nm)/BL(10 nm)/BPhen(8 nm)/Ag(100 nm), where as 'BL' are denoted the materials used as buffer layers (Rubrene, C545T, DBzA, TCTA, TPBA, TBPe, TPBI). The area of the devices was 6.44 mm², defined as the overlap between anode and the Ag cathode. All the used materials were purified twice in-house by vacuum gradient sublimation. The solar cells were encapsulated in nitrogen atmosphere with a transparent encapsulation glass, fixed by UV-hardened epoxy glue. Device characteristics from multiple batches are provided in Supplementary Table 7.

**Solar cells characterisation**. *Current voltage (J–V) measurements*: J–V measurements were carried out on encapsulated devices in ambient conditions using a source measurement unit (SMU 2400 Keithley, USA) and a simulated AM1.5G illumination (16S-003-300-AM1.5G sunlight simulator, Solar Light Co., USA). Masks were used to minimise edge effects and to define an exact photoactive area (2.78 mm²). A silicon (Si) photodiode (Hamamatsu S1337), calibrated by Fraunhofer ISE, was used as reference. Spectral mismatch was taken into account during the measurement.

*EQE measurements*: EQE measurements were performed using a xenon lamp (Oriel Xe Arch-lamp Apex, Newport, USA), a monochromator (Cornerstone 260 1/4 m, Newport, USA), an optical chopper and a lock-in amplifier (SR 7265, Signal Recovery, USA). The EQE of the OSCs was measured with an aperture mask (2.78 mm²) and without bias light. A Si photodiode (Hamamatsu S1337), calibrated at Fraunhofer ISE, was used as reference.

*Sensitive EQE (sEQE) measurements*: The light of a quartz halogen lamp (50 W) was chopped at 140 Hz and coupled into a monochromator (Newport Cornerstone 260 1/4m, USA). The resulting monochromatic light was focused onto the OSC, its current at short-circuit conditions was fed to a current pre-amplifier before it was analysed with a lock-in amplifier (Signal Recovery 7280 DSP, USA). The time constant of the lock-in amplifier was set as 500 ms and the amplification of the pre-amplifier was increased to resolve low photocurrents. The sEQE was determined by dividing the photocurrent of the OSC by the flux of incoming photons, which is obtained with calibrated Si and indium–gallium–arsenide photodiodes.

**Spectroscopy characterisation**. *UV–Vis absorption spectroscopy*: A PerkinElmer Lambda 25 spectrometer was used to carry out UV–Vis absorbance for thin film samples.

*Steady state PL spectroscopy*: Steady-state PL spectra were measured on a Fluorolog-3 spectrofluorometer (FL 3–22, Horiba Jobin Yvon). All the samples were excited at 450 nm with a slit width of 5 nm. The emitted photons were collected with the front-face geometry with a slit width of 5 nm.

*Time-correlated single photon counting (TCSPC)*: The DeltaFlex TCSPC system (Horiba Scientific) was used to measure the time-resolved PL kinetics of thin film samples. Samples were excited by a nano-LED at 404 nm. Photons were detected at 580 nm with a picosecond photon detector. The instrument temporal resolution is within 200 ps.

*TA spectroscopy measurements (TAS)*: A broadband femtosecond TA spectrometer Helios (Spectra Physics, Newport Corp.) was used to measure TAS for thin films and devices. Laser pulses (800 nm, 100 fs pulse duration) were generated using a 1 kHz Ti:sapphire regenerative amplifier (Solstice, Spectra Physics). One portion of the 800-nm pulses was directed to an optical parametric amplifier (TOPAS) to generate the visible pump pulses at 450 nm. The rest of the 800 nm pulses was routed onto a mechanical delay stage (6 ns time window) and was directed through a sapphire crystal to generate a white light probe ranging from 400–900 nm in the visible to near-infra-red region. The pump and probe beams were focused onto the same spot on the samples. During the measurements, the thin film samples were kept in a quartz cuvette under continuous nitrogen flow and the device samples were encapsulated.

*EA measurements*: EA was measured on a full OPV device in the TAS set up with a reflectance geometry. During the measurement, a square wave bias (5 V, 500 Hz, 100 μs high-period) synced with the pulsed laser was applied to the device using a digital delay generator (Stanford Research Systems DG645). The difference in reflectance between bias on and off was measured with various bias voltages, yielding the EA spectrum under different electric fields.

*Ambient photoemission spectroscopy (APS) and surface photovoltage (SPV) measurements*: APS04 (KP Technology) system was employed to carry out the APS and SPV measurements. The measurement sequence is Kelvin probe, SPV, and then APS. The Fermi level measurement uses the off-null Kelvin probe technique where the 2 mm gold tip is applied with oscillating positive or negative 7 V. The contact potential difference measured between the tip to the sample determines the dark work function of the sample. For SPV measurements, samples are kept in dark for at least 10 min to reach equilibrium condition monitored by Kelvin probe. The white light source from quartz tungsten halogen lamps are switched on for 100 s after the first 20 s dark condition. The decay is also recorded for another 150 s after illumination. To measure the HOMO levels, a UV light (−4.4 to −6.2 eV) illuminated the samples to generate photoemission of electrons. The photoemitted electrons interact with air molecules and generate radicals, which are further collected by the positively biased tip. The HOMO levels were subtracted linearly from the cube root photoemission intensity to the baseline.

**Morphology characterisation**. *Grazing-incidence wide-angle X-ray scattering (GIWAXS)*: GIWAXS measurements were performed at the XRD1 beamline at the ELETTRA synchrotron in Trieste. The thin films were illuminated under a grazing angle of 0.13° and the diffraction pattern was recorded with PILATUS 2M area detector, which was placed 35 cm behind the samples. The measurements were performed at a beam energy of 12.4 keV. The data were analysed with the WxDiff software (c S.C.B.M.).

**Simulations**. Theoretical calculations were performed for a 2D-infinite interface between standing and lying a-6T domains. The interface morphology has been built from the experimental crystal structure[60] and relaxed with classical simulations (Supplementary Fig. 12) based on a validated force field[61] performed with the NAMD code[62]. Hybrid quantum/classical (QM/MM) $GW$[53] and Bethe–Salpeter[57] many-body electronic structure calculations were performed with the FIESTA package. These calculations considered up to six molecules in the QM region, embedded into the MM electrostatic and polarisable crystalline environment described with the charge response model in the MESCal code implementation[63]. The starting Kohn–Sham orbitals were obtained with the gap-tuned density functional ωB97X ($\omega = 0.141$ Bohr$^{-1}$) and the cc-pVTZ basis, which ensures accurate energy levels with single-iteration $G_0W_0$ calculations (Supplementary Fig. 13 and Supplementary Table 4). Bethe–Salpeter calculations were performed beyond the Tamm–Dancoff approximation, including all electron–hole transitions up to 10 eV (Supplementary Tables 5 and 6). Full details on electronic structure calculations are provided given in Supplementary Tables 3–6 and Supplementary Figs. 13–19.

**Reporting summary**. Further information on research design is available in the Nature Research Reporting Summary linked to this article.

## Data availability

The datasets generated during and/or analysed during the current study are available from the corresponding author on reasonable request.

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

## Acknowledgements

The authors thank Maxim Pschenitchnikov for useful discussion. This work was supported by the German Federal Ministry of Education and Research (BMBF) through the Innoprofile project 'Organische p-i-n Bauelemente2.2' (FKZ 03IPT602X), UKRI Global Challenge Research Fund project SUNRISE (EP/P032591/1). Y.-C.C. acknowledges the President's PhD Scholarship funding by Imperial College London. A.A.B. is a Royal Society University Research Fellow. J.-S.K. and J.R.D. acknowledge the Global Research Laboratory Program of the National Research Foundation (NRF) funded by the Ministry of Science, ICT & Future Planning (NRF-2017K1A1A2 013153). This project has also received funding from the European Research Council (ERC) under the European Union's Horizon 2020 research and innovation programme (grant agreements 639750 and 864625). The work in Mons was supported by the European Union's Horizon 2020 research and innovation programme under Marie Sklodowska Curie Grant agreement No. 722651 (SEPOMO). Computational resources were provided by the French GENCI-TGCC infrastructure, by the Belgian Consortium des Équipements de Calcul Intensif (CÉCI), funded by the Fonds de la Recherche Scientifiscs de Belgique (F.R.S.-FNRS) under Grant No. 2.5020.11, and by the Tier-1 supercomputer of the Fedération Wallonie-Bruxelles, infrastructure funded by the Walloon Region under Grant Agreement No. 1117545. D.B. is a FNRS Research Director. We acknowledge Elettra Sincrotrone Trieste for providing access to its synchrotron radiation facilities and we thank Luisa Barba for assistance in using beamline XRD1.

## Author contributions

Y.D., V.C.N., J.R.D., A.A.B. and K.V. designed the experiments. V.C.N. prepared films and fabricated and optimised the photovoltaic devices. Y.D. performed the absorbance, steady-state and time-resolved PL measurements and transient absorption studies and their analysis. Y.D. and X.Z. carried out EA measurements. F.T. and S.C.B.M. performed and analysed the GIWAXS measurements. J.K. and J.B. measured and analysed the sensitive EQE spectra. V.C.N. performed the J–V and EQE measurements. Y.-C.C. performed the APS measurements. J.-S.K., D.S., S.C.B.M., J.R.D., A.A.B. and K.V. supervised their team members involved in this project. J.R.D, A.A.B., J.-S.K. and K.V. supervised the overall project. G.D., G.L., J.L., L.M., X.B. and D.B. designed and performed the theoretical and computational analysis. Y.D. and V.C.N. wrote the manuscript. All authors contributed to the critical analysis of the results and to the revision of the manuscript.

## Competing interests

The authors declare no competing interests.
