## [Peer Review File · Nature Communications]

Reviewers' Comments:

Reviewer #1:

Remarks to the Author:

Single component organic solar cells (SCOSC) are the oldest topic in this area, but they are very important to understand the charge transfer and separation mechanism which is not fully clear. The manuscript by Vandewal et al. report efficient charge generation in single component organic solar cells based on molecular electrostatic engineering. Efficient photocurrent generation in α -6T based SCOSC, in the absence of an electron accepting material, reaching an external quantum efficiency of 44% and a VOC of 1.61 V. The result is interesting and important to further clarify the charge-separation mechanism, as well as prepare the higher efficiency SCOSC. My comments are listed as following for further revising the manuscript.

- (1) The Voc of devices with different buffer layers are quite different. Since the charge generation are mainly happened in the α -6T, the reason for the Voc differences need to be investigated.
- (2) The relative strong EA absorption features at the early time is quite interesting, but also is difficult to understand. A more detailed explanation is needed.
- (3) The absorption spectra for α -6T thin films with different molecular orientations is obviously different, while the PL spectra for mixed oriented film is quite similar with the film with lying molecules. The reason behind that need to be explained.
- (4) There is a small peak for mixed oriented film, where this peak come from?
- (5) Is that possible to analyze the GIWAXS of mixed oriented film quantitatively to give the ratio of standing and lying molecules?
- (6) Some closely related manuscript need to be cited, such as Y.J. Zhang et al. Sol. RRL, 2020, 4: 1900580. doi:10.1002/solr.201900580

Reviewer #2:

Remarks to the Author:

A novel kind of single component OSCs is fabricated by using the common material α -sexithiophene (α -6T) in this article and the highest PCE of these devices is 2.9%, which is quite encouraging. Transit absorption spectroscopy demonstrated that charge separation happens in the bulk of α -6T. With the help of other measurements and calculations, the authors illustrated that different crystallite orientations induce an interfacial energy offset of 0.4 eV, which is beneficial to generate free carriers. However, several problems should be tackled before accepting it.

1. From the device fabrication section, it has been pointed out that CuI was firstly deposited on the ITO to control the orientation of α -6T. Apart from that, whether the CuI works as an electrode modifier? In other words, whether the CuI can help extract holes or electrons from the active layer? Similarly, there are buffer layers between active layers and the top electrodes. However, BPhen seems to be the necessary layer. What are the device performances without inserting BPhen between BL and top electrode?
2. For the illustration of the device structure in Fig 1a, CuI should be added. The thickness of α -6T should be 80nm according to the fabrication section. And the thickness unit (nm) should better be directly added instead of being denoted in the caption.
3. The authors employed various characterization methods to prove the excitons separate at the interfaces of lying and standing α -6T. However, devices with only lying α -6T or standing α -6T were not fabricated. I think the data is quite important because it can be rationally inferred that if the exciton dissociation only occurs at the interfaces, the devices based on α -6T with a single orientation will not work.
4. Page 6 and page 7, "Except for a shorter exciton lifetime, all the spectral features are nearly identical with those in pristine α -6T thin films on ITO". However, low and high excitation fluences was used to investigated the exciton decay time of the pristine α -6T thin film and the α -6T based OSCs, respectively. As the excitation fluence affects the exciton decay time, the authors should use the same excitation fluence for two blends.
5. In page 7, "In the lying orientation, the absorbance is maximized due to the parallel alignment

between the transition dipole moment of the molecule and the electric field vector of incident light (Fig. 3a).” Explanations or references are needed.

6. What are the electron and hole mobilities of standing and lying α -6T, respectively? Because α -6T is a donor, how do the electrons transport from interfaces to electrodes through the α -6T?

7. In page 9, “Additionally, to the Bragg-rods visible in-plane ($q_{xy} = 1.31, 1.60$ and 1.91 \AA^{-1} peaks can be seen at positions corresponding to the lying molecules, further showing that mixed molecular orientations are present in these films.”. These peaks are also visible in the film of standing molecules (Fig. 3d). Maybe there is something wrong about this.

8. From a single molecular view, the conjugated plane of the α -6T has a higher electron density than the side profile due to the orientation of pz orbitals. Therefore, it seems that standing α -6T has lower potentials than lying α -6T?

9. The authors attribute the charge separation to the molecular electrostatic engineering in the title. However, the experimental and computational data implies the energetic offsets are the original reasons. Furthermore, the energetic offsets are caused by the different molecular orientations of α -6T. Therefore, to make the title clear and specific, I think the molecular electrostatic engineering in the title should be replaced by other words.

Reviewer #3:

Remarks to the Author:

The solar cells of the current work are not single component as they contain a number of different layers. Assuming that the α -6T is the layer in which light is generated then the devices can be defined as homojunctions. The title could be changed to highlight this point. The authors provide sufficient evidence that the different molecular orientation in the films can lead to ionization potentials that have a difference of 0.4 eV. Assuming that the optical gap is similar for both orientations this would mean that both photoinduced hole and electron transfer could in principle happen. One thing that is not clear from the transient absorption measurements is whether negative and positive polarons are being formed on the α -6T – would they have the same energies? Is there another feature at longer wavelength? The devices are stated to have an α -6T active layer 70 nm thick but the experimental seems to imply a thickness of 20 nm and 60 nm – which is correct? To assist in proving that charge generation only occurs in the bulk of the α -6T active layer it is recommended that the authors model the position of exciton generation in the devices. Given the relative thicknesses of the layers and the extinction coefficients of the materials, are charges being generated at the buffer layer/BCP interface? The authors only show PL data for the α -6T/rubrene combination but should show it for the all complete stacks and demonstrate that there is not a wavelength or thickness dependence. A curious result is the fact that the device with the TCTA buffer layer also has a low energy loss. It is generally accepted that TCTA does not transport electrons – is tunneling expected here? An open circuit voltage of 1.61 V is impressive as is the overall efficiency of the devices. However, there are still substantial losses in V_{oc} and it would be interesting if the authors could provide evidence for where these losses come from and indeed why the PCE is not higher. For example, is there an imbalance in charge mobility? What is the light intensity dependence of the IQE? Finally, there are no device statistics – these should be included. In conclusion, the results are worthy of publication once the extra information is provided. If the results can be extended to other materials it could prove to be an important step forward for organic solar cells.

We would like to thank the reviewers for their constructive comments on our manuscript. Below, we address all their comments in a point-by-point response, along with details on edits made to the manuscript. We believe these revisions now address all issues raised by the reviewers and hope that they find this revised manuscript appropriate for publication in Nature Communications.

Reviewer #1 (Remarks to the Author):

Single component organic solar cells (SCOSC) are the oldest topic in this area, but they are very important to understand the charge transfer and separation mechanism which is not fully clear. The manuscript by Vandewal et al. report efficient charge generation in single component organic solar cells based on molecular electrostatic engineering. Efficient photocurrent generation in α -6T based SCOSC, in the absence of an electron accepting material, reaching an external quantum efficiency of 44% and a V_{OC} of 1.61 V. The result is interesting and important to further clarify the charge-separation mechanism, as well as prepare the higher efficiency SCOSC. My comments are listed as following for further revising the manuscript.

Authors' Response:

We appreciate that Reviewer #1 stresses the importance of understanding charge transfer and separation mechanisms in single component organic solar cells (SCOSC) and assesses our results interesting and important in clarifying these mechanisms and in providing future design guidelines for higher efficiencies. In light of the first comment made by reviewer #3, we have decided that "homojunction organic solar cells (HOSC)" better indicate the type of devices dealt within this work. From now on, we therefore refer to these devices with the abbreviation HOSC instead of SCOSC.

(1) The V_{OC} of devices with different buffer layers are quite different. Since the charge generation are mainly happened in the α -6T, the reason for the V_{OC} differences need to be investigated.

Authors' Response:

This is indeed an interesting point, which has not been sufficiently elaborated in the previous version of the manuscript. We thank the reviewer for giving us the opportunity to do so through this revision round. The fact that for various buffer layers (BLs), the sensitively measured EQE spectra (Supplementary Figure S2) are identical in the subgap region, indicates that the addition of the BL materials does not create an interface with a much lower energy than the α -6T gap. This only occurs for the α -6T/BPhen interface, leading to the lowest V_{OC} (1.25 V, as compared to 1.61 V). For the higher V_{OC} devices, no subgap states related to the other α -6T/BL interfaces are detected, which could in principle reduce V_{OC} . The insertion of a BL material between α -6T and BPhen targeted exactly on avoiding the formation of such a low energy interface.

To gain more insight into the V_{OC} losses, we performed a voltage loss analysis and calculated the maximum voltage at the radiative limit (V_{rad}) for the tested devices. We see that, except for the α -6T/BPhen device, the V_{rad} is almost identical for all the other devices, varying between 1.87 V and 1.89 V, implying that the lower V_{OC} values are due to increased non-radiative recombination ($\Delta V_{non-rad} = V_{rad} - V_{OC}$) in those devices. This increased non-radiative recombination of free charge carriers is also reflected in a reduction of the FF (Table 1). Since the scope of the current manuscript focuses on the fundamental understanding of the efficient charge generation mechanism in α -6T, we consider that identifying the exact role of the buffer layer materials on free carrier recombination in our devices would not alter the main outcome of this work.

Based on the performed voltage loss analysis, we have updated the Supplementary Table S2, which now shows the V_{rad} and $\Delta V_{non-rad}$ values. Moreover, we have updated the main text (page 4) as shown below (changes are marked in yellow):

“...the use of extra BLs improves significantly the V_{OC} (up to 1.61 V) compared to the optical gap of α -6T (2.33 eV, Supplementary Figure S1), mainly due to a reduction of the non-radiative recombination in those devices, leading to total energy losses of 0.72 eV in the device with Rubrene and TPBA (Supplementary Table S2).”

Table S2. Energy and voltage losses in the devices employing α -6T and various buffer layers. The voltage at the radiative limit (V_{rad}) is calculated based on the EQE spectra of the devices shown in Figure S2, using the method outlined in ref. 16. The non-radiative voltage losses ($\Delta V_{non-rad}$) are estimated by subtracting the open-circuit voltage of the devices (V_{OC}) from V_{rad} . The total energy losses are estimated by subtracting the V_{OC} from the optical gap (E_g) of α -6T (2.33 eV). The lowest energy losses (0.72 eV) are observed for the device with rubrene, which provides the highest V_{OC} (1.61 V).

Device structure	V_{OC} (V)	V_{rad} (V)	$\Delta V_{non-rad}$	$E_g - qV_{OC}$ (eV)
ITO / α -6T / BPhen / Ag	1.25	1.58	0.33	1.08
ITO / α -6T / Rubrene / BPhen / Ag	1.61	1.87	0.26	0.72
ITO / α -6T / C545T / BPhen / Ag	1.46	1.89	0.40	0.87
ITO / α -6T / DBzA / BPhen / Ag	1.57	1.88	0.31	0.76
ITO / α -6T / TCTA / BPhen / Ag	1.57	1.88	0.31	0.76
ITO / α -6T / TPBA / BPhen / Ag	1.61	1.88	0.27	0.72
ITO / α -6T / TBPe / BPhen / Ag	1.41	1.88	0.47	0.92
ITO / α -6T / TPBI / BPhen / Ag	1.50	1.88	0.38	0.83

(2) The relative strong EA absorption features at the early time is quite interesting, but also is difficult to understand. A more detailed explanation is needed.

Authors' Response:

The prompt rise of the strong EA feature is in line with the ultrafast charge carrier generation as suggested by the prompt rise of the kinetics at 770 nm. The reason that it is very strong at the early time is that the initially generated excitons have a high degree of charge-transfer (CT) character, most likely associated with the hybridisation between exciton and CT states, as also suggested by our electronic structure calculations. However, we are also aware that the EA absorption feature is partially masked by the ground state bleaching (515-535 nm) and the photoinduced absorption of excitons (495-515 nm).

A similar observation of the prompt and strong EA rise has also been reported in literature by several groups including Banerji et al., Moser et al. [Causa', M. et al. The fate of electron-hole pairs in polymer:fullerene blends for organic photovoltaics. Nat Commun 7, 12556 (2016); Devižis et al. Dissociation of Charge Transfer States and Carrier Separation in Bilayer Organic Solar Cells: A Time-Resolved Electroabsorption Spectroscopy Study. J. Am. Chem. Soc. 2015, 137, 8192–8198].

Based on this comment, we have added the following sentences and relevant references on page 5 as shown below to address this phenomenon (changes are marked in yellow):

“We independently measured the EA of the α -6T HOSC (Fig. S5) and validated the resemblance between the EA and TA spectra, which further suggests that the strong signal at 525 nm and 508 nm in TA arises from the initially generated excitons having a high-degree of charge transfer character, most likely associated with the hybridisation between exciton and CT states (see further discussions below). The observation of similar EA signals has been reported for D-A organic solar cells.^{25,26}”

(3) The absorption spectra for α -6T thin films with different molecular orientations is obviously different, while the PL spectra for mixed oriented film is quite similar with the film with lying molecules. The reason behind that need to be explained.

Authors' Response:

As shown in the absorption spectra (Fig. 3a), the main difference among samples with different orientations lie in the high-energy region, whereas the transition near the band edge does not change much. Since PL originates from the lowest excited state, with the excitations undergoing a significant vibrational relaxation, as indicated by the large Stokes shift. Moreover, the emission setup detects photons emitted over a large range of angles. Therefore, the emission spectra for the various orientations do not drastically differ in shape, only in intensity.

Based on this, we have made the following change in the main text (page 5) as shown below (changes are marked in yellow):

“The mixed orientation lies in between the two, indicating the co-existence of both orientations (see GIWAXS data in Fig. 3d-f, discussed in detail below). While the absorption spectra vary in shape and intensity for differently oriented films, the photoluminescence (PL) spectra retain the same shape since they originate from the lowest excited state after the excitations undergoing a significant vibrational relaxation. However, PL and time correlated

single photon counting (TCSPC) show a quenching and a faster photoluminescence decay in the mixed film in comparison with the other two orientations (Fig. 3b,c), implying that an enhanced exciton quenching mechanism is present in the mixed film containing both standing and lying α -6T phases.”

(4) There is a small peak for mixed oriented film, where this peak come from?

Authors' Response:

It is unclear to us which peak the referee refers to: If it is in the TCSPC data, the peak at 1.2 ns is coming from the instrument response which is only partially covered by the “mixed” sample signal, which is weaker than the others. Actually, a shoulder is visible also in the “lying” signal. The “prompt” decay in Figure 3c indeed represents the instrument response function measured with a glass substrate.

Based on this comment of the Reviewer, we feel that it is worth clarifying this better in the caption of Fig. 3, as shown below (changes are marked in yellow):

Fig. 3 Morphological and spectroscopic characterisation for the orientation of α -6T thin films. **a**, Absorption spectra for α -6T thin films with different molecular orientations where the lying molecules show the highest absorbance and the mixed orientation film lies in between the lying and standing samples. **b**, Photoluminescence (PL) spectra normalised with the absorbance at the excitation wavelength of 450 nm for α -6T thin films with different molecular orientations where the PL quenching is observed in the mixed molecules. **c**, Time-correlated single photon counting for α -6T thin films with different molecular orientations (standing, mixed and lying) revealing a faster PL decay in the presence of both standing and lying orientations. The grey dots indicate the prompt decay (instrument response function), which in the case of the mixed and lying orientations induces an artefact at 1.2 ns. **d-f**, Grazing-incidence wide-angle x-ray scattering (GIWAXS) diffraction images showing that the

molecular orientation of α -6T thin films can be tuned with specific processing conditions. The peaks indicated by red and blue circles originate from crystallites with standing and lying molecules, respectively. **g-i**, schematic morphology for α -6T thin films with different molecular orientations (standing, mixed and lying).

(5) Is that possible to analyze the GIWAXS of mixed oriented film quantitatively to give the ratio of standing and lying molecules?

Authors' Response:

In theory this is possible and is reported for polymers e.g. to determine face-on and edge-on populations in P3HT films. For this, the peak intensities for specific (hkl) reflections at different azimuth angles χ are measured and corrected and then compared. For a polymer this is comparably easy as it is (in the best case) only one broad ring along which one would have to integrate. In our case, the diffraction images are more complicated. We would have to find two peaks on one azimuthal (the scattering vector q is the same but the azimuthal angle χ goes from 0° to 90°) with the same (hkl) and no other peaks close by to be able to reliably extract the intensity of these peaks. Nevertheless, the large amount of peaks visible and the large width of the peaks in the 2D diffraction image of the mixed film, makes it unfortunately challenging to extract a reliable value for the ratio of the two populations. Additional problems for extracting the intensity of the peaks are blind spots on the detector which is problematic for the peaks in (H00) Bragg rod ($q_z=0$).

(6) Some closely related manuscript need to be cited, such as Y.J. Zhang et al. Sol. RRL, 2020, 4: 1900580. doi:10.1002/solr.201900580

Authors' Response:

We have now added this reference on page 3 (ref. 14). It is worth noting that the molecule used in this case is fundamentally different from that in our work, where the former possesses an A-D-A type of chemical structure but our α -6T molecule is only considered as a donor. Nevertheless, we would like to thank the Reviewer for this suggestion.

Reviewer #2 (Remarks to the Author):

A novel kind of single component OSCs is fabricated by using the common material α -sexithiophene (α -6T) in this article and the highest PCE of these devices is 2.9%, which is quite encouraging. Transit absorption spectroscopy demonstrated that charge separation happens in the bulk of α -6T. With the help of other measurements and calculations, the authors illustrated that different crystallite orientations induce an interfacial energy offset of 0.4 eV, which is beneficial to generate free carriers. However, several problems should be tackled before accepting it.

Authors' Response:

We appreciate that the Reviewer #2 thinks the device performance is very encouraging. We have addressed his/her constructive comments below.

1. From the device fabrication section, it has been pointed out that Cul was firstly deposited on the ITO to control the orientation of α -6T. Apart from that, whether the Cul works as an electrode modifier? In other words, whether the Cul can help extract holes or electrons from the active layer? Similarly, there are buffer layers between active layers and the top electrodes. However, BPhen seems to be the necessary layer. What are the device performances without inserting BPhen between BL and top electrode?

Authors' Response:

We would like to clarify that Cul is not included in the shown devices, but only in the thin-film samples for the 'lying' orientation, which were used for the TA, APS and GIWAXS measurements. Nevertheless, we have realized that the section that the Reviewer refers to, about the fabrication of thin-film samples in Methods, had mistakenly the title '*Thin film and device preparation*', although the fabrication of devices is described in the next section '*Device fabrication*'. We would like to apologize for the confusion and thank the Reviewer for noticing the inconsistency. In the revised version of the manuscript, the section is renamed to '*Thin film samples fabrication*'.

Since we have not used Cul in our devices, we have not investigated its effects on charge extraction, although recent reports such as [Peng, Y. et al. Efficient organic solar cells using copper(I) iodide (Cul) hole transport layers, Applied Physics Letters 106, 243302 (2015)] show that Cul would work efficiently as hole transport layer in organic solar cells. On the other hand, BPhen was indeed used in all our devices since it is a well-known ETL, which blocks holes efficiently.

As we mentioned in the first comment of Reviewer #1, the scope of the current manuscript focuses on the understanding of the efficient charge generation mechanism in α -6T. We agree with Reviewer #2 that understanding the exact role of BPhen in the shown devices would be interesting, however, as the buffer layer materials in our devices will mainly affect the recombination of free carriers (at the contact), it is not expected that there is an impact on the conclusions made in this work regarding the charge generation mechanism in α -6T.

2. For the illustration of the device structure in Fig 1a, CuI should be added. The thickness of α -6T should be 80nm according to the fabrication section. And the thickness unit (nm) should better be directly added instead of being denoted in the caption.

Authors' Response:

As discussed in the previous comment, CuI was used only for fabricating the thin-film samples with the 'lying' orientation for the TA, GIWAXS and APS measurements. In the investigated devices the structure is as shown in Figure 1a with 70 nm of α -6T deposited on ITO, then 10 nm of BL and 8 nm of BPhen are deposited sequentially before 100 nm of Ag are deposited as top electrode.

We have added the unit in the Fig. 1a (on page 4). Furthermore, we have added the full device structure of the investigated devices in the 'Device fabrication' section in Methods (on page 14, changes highlighted in yellow):

"...Every investigated device was bottom illuminated, employing indium tin oxide (ITO) as the anode and a 100 nm thick Ag as the cathode. The full device structure of the investigated devices is as follows: ITO/ α -6T(70 nm)/BL(10 nm)/BPhen(8 nm)/Ag(100 nm), whereas 'BL' are denoted the materials used as buffer layers (Rubrene, C545T, DBzA, TCTA, TPBA, TBPe, TPBI)."

3. The authors employed various characterization methods to prove the excitons separate at the interfaces of lying and standing α -6T. However, devices with only lying α -6T or standing α -6T were not fabricated. I think the data is quite important because it can be rationally inferred that if the exciton dissociation only occurs at the interfaces, the devices based on α -6T with a single orientation will not work.

Authors' Response:

We agree with the Referee that, having attributed the exciton dissociation and charge generation mechanisms to the interfaces between crystallites with different orientations, it would indeed be interesting to study devices where such interfaces are absent. Such an appealing possibility is unfortunately not applicable to our case, mostly because of the two following reasons:

Firstly, it is practically impossible to fabricate devices or thin films with either exclusively 'standing' or 'lying' orientation. There are always misoriented domains present.

In second instance, the fabrication of non-functional devices with only one molecular orientation would not necessarily strengthen our conclusions, since a reduction in the photocurrent or EQE could also originate from issues with charge extraction or transport in such devices. In short, there are many reasons related to device processing which could result in non-functional devices, which makes it hard to distinguish the charge generation process from other factors diminishing device performance.

Therefore, we believe that the practical approach we have adopted: isolating the most important component of our devices, i.e. the neat α -6T layer, and probing the charge generation mechanisms in films containing various fractions of orientations with TA, while excluding any external influence that can interfere with our observations, is the safest way to support our interpretation for the unprecedented performances for a HOSC.

Furthermore, theoretical calculations provide a neat and robust explanation for the charge generation at boundaries between 6T domains with different orientations. On the other hand, calculations allow us to conclude that CT excitons in a single domain would be ~ 0.2 eV higher in energy (preventing the hybridization with low-energy Frenkel molecular excitons) and would present a ~ 0.2 eV larger electron-hole binding (hampering charge separation). Now, these considerations have been included in the SI as Note S3, expanding the discussion of data in Table S3.

4. Page 6 and page 7, "Except for a shorter exciton lifetime, all the spectral features are nearly identical with those in pristine α -6T thin films on ITO". However, low and high excitation fluences was used to investigated the exciton decay time of the pristine α -6T thin film and the α -6T based OSCs, respectively. As the excitation fluence affects the exciton decay time, the authors should use the same excitation fluence for two blends.

Authors' Response:

We would like to thank the Reviewer for pointing this out. We apologise for the inaccurate description regarding the exciton lifetime in the main text. Indeed, the excitation fluence can affect the exciton decay time. We have carefully compared our data and elucidated that the exciton lifetimes in the device and the film show good agreement at the same excitation fluence. The only difference is the charge kinetics, where a slightly faster recombination is observed in the device compared to the film.

As shown in Supplementary Figure S8 (kinetics for excitons in the thin film), the kinetics have almost the same decay rate when excited with an excitation fluence of either 10 or 5 $\mu\text{J cm}^{-2}$, indicating that exciton-exciton annihilation effect is negligible when the fluence reaches 10 $\mu\text{J cm}^{-2}$ or below. We have therefore measured the two samples (pristine α -6T thin film and the α -6T based OSC device) under the same excitation fluence of 10 $\mu\text{J cm}^{-2}$. Shown below is the normalised exciton decay dynamics (probed at 593 nm) for the film and the device. The decay half-life times for the two samples are in reasonable agreement, although the film data is noisier than the device data.

The slightly faster decay of charges in the device compared to that in the film is likely related to the charge extraction assisted by the electrodes. Since the current manuscript focuses on the understanding of the charge generation mechanism in α -6T, we believe that the electrode effect on charge recombination would not change the main conclusion from this work.

To clarify this point, we have added the following figure to the Supplementary Information as Figure S9 to highlight the agreement in exciton lifetimes between the film and the device.

“Figure S9. Normalised transient absorption kinetics comparison for the excitons in α -6T thin film with device. The kinetics for excitons probed at 593 nm show a similar half-lifetime in both samples. Both samples were excited at $10 \mu\text{J cm}^{-2}$.”

We’ve also made the following change in the main text (on page 7):

“Turning to the α -6T based HOSC with rubrene as BL, similar exciton decay lifetimes were observed (Fig. S6, S8). Except for a slightly faster charge recombination kinetics, all the spectral features are nearly identical with those in pristine α -6T thin films on ITO, indicating that efficient free charge carrier generation is not only present in devices but also in α -6T thin films.”

5. In page 7, “In the lying orientation, the absorbance is maximized due to the parallel alignment between the transition dipole moment of the molecule and the electric field vector of incident light (Fig. 3a).” Explanations or references are needed.

Authors’ Response:

It has been reported in the literature that transition dipole moments for electronic excitation are generally aligned with the long axis of the π -conjugated plane of the molecules and the electric field of the light and the dipoles couple most strongly. [Nakano et al. Organic Planar Heterojunctions: From Models for Interfaces in Bulk Heterojunctions to High-Performance Solar Cells. Adv. Mater. 2017, 29, 160326) Specifically, in the case of α -6T molecules, it is well known that the main optical transition dipole aligns parallel to the long molecular axis. [Taliani, C. & Blinov, L. M. The electronic structure of solid α -sexithiophene. Adv. Mater. 8, 353–359 (1996), Ref.24; Kouki et al. Experimental determination of excitonic levels in α -oligothiophenes. J. Chem. Phys. 113, 385 (2000)]

To clarify this, we have added these two references, as ref. 46 and 47, at the mentioned sentence (on page 8):

“...In the lying orientation, the absorbance is maximised due to the parallel alignment between the transition dipole moment of the molecule and the electric field vector of incident light (Fig. 3a).^{46,47}”

6. What are the electron and hole mobilities of standing and lying α -6T, respectively? Because α -6T is a donor, how do the electrons transport from interfaces to electrodes through the α -6T?

Authors' Response:

In our HOSCs, α -6T behaves indeed both as a hole (standing domains, donor) and electron (lying domains, acceptor) transporting material.

Hole mobilities in the 10^{-1} - 10^{-3} cm^2/Vs range have been reported for α -6T crystals and ordered films (see Fichou, J. Mater. Chem. 10, 571, 2000; and references therein). We are not aware of any measurement of the electron mobility for α -6T, yet it is expected to be similar to hole mobility, if one is able to obtain efficient injection at the electrodes (which cannot happen for instance with the Au electrodes commonly used in the literature). Mobilities up to $0.02 \text{ cm}^2/\text{Vs}$ were reported for 6T derivatives with fluorinated side chains (Facchetti et al., Angew. Chem. 112, 4721, 2000).

We emphasize that charge carriers mobilities do not represent a major hurdle for solar cell applications (while this is the case for transistors), and the values given above are larger than in many disordered bulk heterojunctions and do not preclude efficient OPV performances. The measurements of hole and electron mobilities in films of different or coexisting orientations goes beyond the scopes of this work, and certainly applying the concepts of molecular electrostatic engineering to materials with balanced electron and hole mobility represents an interesting perspective for future work.

Nevertheless, in order to clarify the role of 6T as donor or acceptor, depending on its orientation, we have added the following to the main text:

*“...As sketched in Fig. 4d, standing and lying α -6T molecules behave therefore as the electron donating and the accepting components of a conventional organic heterojunction, consistent with the efficient charge generation observed in our TA data above. **In a working device consisting of such components, holes and electrons transport takes place in the domains with standing and lying orientations, respectively.**”*

7. In page 9, “Additionally, to the Bragg-rods visible in-plane ($q_{xy} = 1.31, 1.60$ and 1.91 \AA^{-1} peaks can be seen at positions corresponding to the lying molecules, further showing that mixed molecular orientations are present in these films.”. These peaks are also visible in the film of standing molecules (Fig. 3d). Maybe there is something wrong about this.

Authors' Response:

The diffraction image for the films deposited at room temperature with a high evaporation rate, shown in Fig. 3e, is a superposition of the diffraction images of the films with lying and standing orientation, shown in Fig. 3d and f respectively. This is clearly visible by the existence of the lamellar peaks in the out-of-plane direction (red circles, standing orientation)

and the in-plane direction (blue circles, lying orientation). Additionally, several other peaks for each orientation are observable, which are indicated by red and blue circles depending on whether they originate from the standing or lying molecules. This shows that both molecular orientations are present in the mixed films.

To clarify this point better, we modified the text in the manuscript as following (changes are marked in yellow):

“...The diffraction image for mixed films deposited at room temperature with a high evaporation rate, shown in Fig. 3e, is a superposition of the diffraction images of the films with lying and standing orientation, shown in Fig. 3d and f respectively. This is clearly visible by the existence of the lamellar peaks in the out-of-plane direction (red circles, standing orientation) and the in-plane direction (blue circles, lying orientation). Additionally, to the Bragg-rods visible in-plane ($q_{xy} = 1.31, 1.60$ and 1.91 \AA^{-1}), which originate from the standing orientation, peaks can be seen at positions corresponding to lying orientations, further showing that both molecular orientations are present in these films.”

8. From a single molecular view, the conjugated plane of the α -6T has a higher electron density than the side profile due to the orientation of p_z orbitals. Therefore, it seems that standing α -6T has lower potentials than lying α -6T?

Authors' Response:

We certainly agree that because of the p_z orbitals spilling out of the molecular plane, the electron density above and below the plane is larger than at the edges. Such anisotropy in the molecular charge density confers to α -6T a pronounced electric quadrupole moment. From an isolated (gas phase) molecule standpoint, standing and lying α -6T are, however, equivalent.

The effect on the energy levels discussed in our work are instead due to intermolecular electrostatic interactions associated with molecular quadrupoles. One needs at least two molecules with different orientations to generate a difference between the energy levels (electron affinities or ionization potentials) of the two molecules. Specifically, the difference in the energy levels between lying and standing domains arises from the superposition of the long-range fields of the molecular quadrupoles in a macroscopic interface, as explained in the manuscript.

To further clarify our message, we have modified the following sentence in the introduction:

“...State-of-the-art calculations based on embedded many-body theories revealed that this energy offset dictated by intermolecular electrostatic interactions persists at standing/lying grain boundaries, where it promotes the formation of low-lying excitations with hybrid intramolecular/CT character.”

9. The authors attribute the charge separation to the molecular electrostatic engineering in the title. However, the experimental and computational data implies the energetic offsets are

the original reasons. Furthermore, the energetic offsets are caused by the different molecular orientations of α -6T. Therefore, to make the title clear and specific, I think the molecular electrostatic engineering in the title should be replaced by other words.

Authors' Response:

We agree with the reviewer that the title can be more specific. We have now changed the title to "Orientation Dependent Molecular Electrostatics Drives Efficient Charge Generation in Homojunction Organic Solar Cells".

Reviewer #3 (Remarks to the Author):

The solar cells of the current work are not single component as they contain a number of different layers. Assuming that the alpha-6T is the layer in which light is generated then the devices can be defined as homojunctions. The title could be changed to highlight this point.

Authors' Response:

We appreciate the reviewer's suggestion on the title and agree that it can be more specific. Strictly speaking, we refer to organic solar cells based on a single-component active layer here. Nevertheless, we have changed the title to "Orientation Dependent Molecular Electrostatics Drives Efficient Charge Generation in Homojunction Organic Solar Cells". This title is indeed more precise and descriptive.

Based on this change we have now updated the whole manuscript by substituting the term "Single component organic solar cell (SCOSC)" with the term "Homojunction organic solar cell (HOSC)"

1. The authors provide sufficient evidence that the different molecular orientation in the films can lead to ionization potentials that have a difference of 0.4 eV. Assuming that the optical gap is similar for both orientations this would mean that both photoinduced hole and electron transfer could in principle happen. One thing that is not clear from the transient absorption measurements is whether negative and positive polarons are being formed on the alpha-6T – would they have the same energies? Is there another feature at longer wavelength?

Authors' Response:

Yes indeed, both photoinduced hole and electron transfer can happen, but we are not able to observe them separately. Both electron and hole transfer will generate both positive and negative polarons. In the transient absorption (TA) measurements we are unable to differentiate between negative and positive polarons. So, we cannot deduce their energies.

We have carried out the TA measurements in the near-Infrared region as shown in the figure below. The kinetics at all probe wavelengths show the same kinetics and was fitted to be 58 ps, which is on the same order of the exciton decay lifetime in the visible region (Fig. 3b). However, we didn't observe any more photoinduced absorption features related to the polarons.

Based on this comment, we have added this figure as Figure S10 in the SI.

“Figure S10. Transient absorption spectroscopy characterisation for the α -6T thin film in the near-Infrared region. a. TA spectra as a function of pump-probe time delay. b. TA kinetics probed at 990-1100 nm. The solid line represents the mono-exponential fitting which gives a lifetime of 58 ps. The sample was excited at 450 nm with an excitation fluence of $10 \mu\text{J cm}^{-2}$.”

2. The devices are stated to have an alpha-6T active layer 70 nm thick but the experimental seems to imply a thickness of 20 nm and 60 nm – which is correct?

Authors' Response:

In alignment with our response the first comment of Reviewer #2, we would like to clarify that the fabrication of thin-film samples in Methods, had mistakenly the title ‘*Thin film and device preparation*’, although the fabrication of devices is described in the next section ‘*Device fabrication*’. We would like to apologize for the confusion. The devices were indeed made with an α -6T layer of 70 nm, and the thin-film samples used for APS had two different thicknesses of 20 nm and 60 nm, for each orientation, in order to check that shift of energetics is not affected by the thickness of the film. The thin-film samples used for TA and GIWAXS had a thickness of 60 nm.

In the revised version of the manuscript, the section related to the fabrication of thin-film samples is now renamed to ‘*Thin film samples fabrication*’. Moreover, the text in this section is further modified to mention the thickness of the samples used in each of the mentioned techniques:

“Thin film samples fabrication: The thin film samples were prepared on glass substrates with a sputtered indium tin oxide (ITO) layer. Thin films of either 20 nm or 60 nm were fabricated for each orientation. Both thicknesses were used for the APS measurements, while GIWAXS and TAS were performed on samples which had a thickness of 60 nm.”

3. To assist in proving that charge generation only occurs in the bulk of the alpha-6T active layer it is recommended that the authors model the position of exciton generation in the devices. Given the relative thicknesses of the layers and the extinction coefficients of the materials, are charges being generated at the buffer layer/BCP interface?

Authors' Response:

We would like to mention that the fact that we observe very similar kinetics (in our TAS measurements) in the α -6T pristine films and devices (Figures 3 and S7), is already a strong proof that charge generation occurs in the bulk of α -6T, and is not dependent to any other interface. Nevertheless, we thank the Reviewer for the suggestion to further complement our results with optical field modelling, in order to show that sunlight absorption occurs mostly in the 6T layer of the devices

We simulated the optical field distribution as a function of wavelength, based on transfer matrix modelling, for two devices employing or not the Rubrene layer between α -6T and BPhen. The full layer sequence used in the simulations is: Air / Glass (1100 μ m) / ITO (90 nm) / α -6T (70 nm) / Rubrene (0 or 10 nm) / BPhen (8 nm) / Ag (100 nm). In the wavelength range from 330 nm to 530 nm, which is relevant for the EQE of the devices (in Figure 1). Photons are mostly absorbed in the α -6T layer for both devices, close to the interface with ITO.

The simulation results are shown below and also included in the SI as Supplementary Figure S3.

“Figure S3. Distribution of optical field in two α -6T based HOSCs employing either BPhen, or Rubrene and BPhen between α -6T and the Ag contact. Simulations were performed based on transfer matrix method and show that for both devices the absorption of photons, relevant for the EQE shown in Figure 1 of the manuscript, occurs mostly in the bulk of α -6T close to the interface with ITO.”

4. The authors only show PL data for the alpha-6T/rubrene combination but should show it for the all complete stacks and demonstrate that there is not a wavelength or thickness dependence.

Authors' Response:

Given that we didn't observe any effect of using rubrene even in the best performing device, we think it is unlikely that other layers would have an impact on charge separation. We did collect some data on the TCSPC for films with various thicknesses. We observe that different thicknesses (20 nm and 60 nm) have a negligible effect on the kinetics of the mixed and lying oriented films. However, we also notice that some possible changes in morphology have complicated the kinetics of the standing oriented films, which is why we excluded these data from the manuscript. Nevertheless, the thickness-dependent TCSPC data is still consistent with the main conclusion of this work – which is the quenching of excitons only occurs in the mixed orientation regardless of the thickness.

5. A curious result is the fact that the device with the TCTA buffer layer also has a low energy loss. It is generally accepted that TCTA does not transport electrons – is tunneling expected here?

Authors' Response:

Indeed, TCTA has exhibited one of the highest V_{OC} 's and lowest voltage losses, in our devices. Moreover, TCTA was the material with the highest LUMO (-2.3 eV), among the materials that we used as buffer layers, and 0.3-0.8 eV higher than that of α -6T (Supplementary Table S1). Although these values were measured with cyclic voltammetry and the LUMO values in solid state may differ, it is clear that a significant energetic barrier should exist at the α -6T/TCTA interface that hinders electron extraction. The low FF (39.4%) that we measured for the α -6T/TCTA supports this assumption (Table 1). As the reviewer suggests, tunnelling could indeed aid charge transport here, considering the low thickness of the TCTA (10 nm), as well as the reported high roughness of the α -6T layer, which can lead to a discontinuous/non-uniformly thick TCTA layer deposited on of it [Cnops, K. et al. 8.4% efficient fullerene-free organic solar cells exploiting long-range exciton energy transfer. Nature Communications 5, (2014)].

Based on this comment, we have updated the main text as below (on page 4):

“...The selection of the BLs adjacent to α -6T was based on their lowest unoccupied molecular orbital (LUMO) energy, being comparable to that of α -6T (Supplementary Table S1), and their use focuses on the improvement of contact selectivity and device performance.^{19,21} **In the case that the LUMO of the BL material (TCTA, for instance) is much higher than that of α -6T, we expect tunnelling to aid electron transport, considering the low thickness of the BL layer (10 nm), as well as the reported high roughness of the α -6T layer,¹⁶ which can lead to a discontinuous/inhomogeneous thick BL layer deposited on it.”**

6. An open circuit voltage of 1.61 V is impressive as is the overall efficiency of the devices. However, there are still substantial losses in Voc and it would be interesting if the authors could provide evidence for where these losses come from and indeed why the PCE is not higher. For example, is there an imbalance in charge mobility? What is the light intensity dependence of the IQE?

Authors' Response:

This is an interesting point which we have elaborated also for the first comment of Reviewer #1. From the voltage loss analysis that we have performed, we see that the differences in the V_{OC} values are due to different non-radiative recombination losses. We attributed those losses to the inefficient extraction of charges, which are connected to potentially imbalanced mobilities. We have not looked deeper into the physics of charge extraction in our devices (also with light intensity dependent measurements), since this is out of the scope of the current manuscript, which focuses on the understanding of the efficient charge generation mechanism in α -6T. The shown devices work just as a demonstration of the effects that we see in the α -6T films.

7. Finally, there are no device statistics – these should be included. In conclusion, the results are worthy of publication once the extra information is provided. If the results can be extended to other materials it could prove to be an important step forward for organic solar cells.

Authors' Response:

We thank the Reviewer for recognizing the importance of our findings. Our work was based on one of the archetypal organic electronics molecules, and we strongly believe that the conclusions drawn from our work are applicable to a broad range of organic materials showing the same morphology characteristics.

The devices shown in this work have been processed by thermal evaporation in vacuum, which is a processing technique known for its high reproducibility. Since the current work doesn't focus on high efficiencies, we did not perform extensive device statistics. Nevertheless, we checked the reproducibility of our results by fabricating at least four devices in each case and selected the best device to be included in Table 1 of the main text. The best device is highlighted in bold in the Table below, which have been included in the SI as Supplementary Table S7.

Device	Materials	J_{sc} (mA cm ⁻²)	V_{oc} (V)	FF (%)	PCE (%)
1	α-6T/BPhen	1.33	1.25	41.4	0.7
2	α -6T/BPhen	1.32	1.25	41.0	0.7
3	α -6T/BPhen	1.33	1.25	41.3	0.7
4	α -6T/BPhen	1.31	1.25	41.1	0.7
1	α-6T/TCTA	3.55	1.57	39.4	2.2
2	α -6T/TCTA	3.56	1.56	39.7	2.2
3	α -6T/TCTA	3.53	1.56	39.6	2.2
4	α -6T/TCTA	3.49	1.57	39.1	2.1
1	α-6T/TBPe	3.00	1.41	42.5	1.8
2	α -6T/TBPe	3.02	1.41	42.0	1.8
3	α -6T/TBPe	3.03	1.41	42.1	1.8
4	α -6T/TBPe	3.00	1.41	41.9	1.8
1	α-6T/C545T	3.82	1.46	34.7	1.9
2	α -6T/C545T	3.80	1.48	33.6	1.9
3	α -6T/C545T	3.79	1.46	33.5	1.8
4	α -6T/C545T	3.81	1.46	33.3	1.8
1	α -6T/DBzA	3.27	1.56	40.6	2.1
2	α-6T/DBzA	3.33	1.57	39.9	2.1
3	α -6T/DBzA	3.27	1.57	39.8	2.0
4	α -6T/DBzA	3.33	1.56	39.4	2.0
1	α-6T/TPBI	3.45	1.50	33.3	1.6
2	α -6T/TPBI	3.43	1.48	32.9	1.6
3	α -6T/TPBI	3.44	1.50	33.0	1.6
4	α -6T/TPBI	3.41	1.47	33.6	1.6
1	α-6T/TPBA	3.61	1.61	47.0	2.8
2	α -6T/TPBA	3.58	1.60	45.5	2.7
3	α -6T/TPBA	3.62	1.61	45.6	2.8
4	α -6T/TPBA	3.63	1.58	46.3	2.7
1	α -6T/Rubrene	3.79	1.61	50.3	2.9
2	α-6T/Rubrene	3.80	1.61	50.2	2.9
3	α -6T/Rubrene	3.78	1.61	50.4	2.9
4	α -6T/Rubrene	3.75	1.61	50.1	2.8

Reviewers' Comments:

Reviewer #1:

Remarks to the Author:

The manuscript has been revised carefully, which could be accepted at its current form.

Reviewer #2:

Remarks to the Author:

The authors have addressed my concerns. I am ok with this revision.

Reviewer #3:

Remarks to the Author:

The authors have clearly worked hard to address the comments of all the reviewers and should be commended for their efforts. There are three outstanding issues that remain to be addressed. First, the authors do really need to address the recombination in the devices. If they are proposing the homojunction route as a way forward for efficient organic solar cells then it is not just the charge generation that is important but also the fate of the generated charges. Knowledge of this is critical. Second, the authors should measure the charge mobilities, electron and hole of the films they are preparing - the assumption that they are essentially the same is bold. Finally, it appears that the devices are all from a single batch with a chip containing four devices. If that is the case then there should be at least two batches produced to prove the statement that the process is reproducible.

Thank you again for submitting your manuscript "Orientation Dependent Molecular Electrostatics Drives Efficient Charge Generation in Homojunction Organic Solar Cells" to Nature Communications. We have now received reports from 3 reviewers and, on the basis of their comments, we have decided to invite a revision of your work for further consideration in our journal. Your revision should address all the points raised by our reviewers (see their reports below). In particular, Reviewer #3 raises a few technical concerns related to charge transport and carrier recombination in the devices, as well as the need to validate the reproducibility of your devices with additional device work. We would like to see your response to these concerns before making a decision on publication.

We are delighted to see that Reviewers #1 and #2 are happy with the revised version of our manuscript. We provide additional experimental data and all the necessary clarifications in order to tackle the remaining concerns. We are confident that the below clarifications, new data, and revisions will satisfy the last three points of Reviewer #3.

Reviewer #1 (Remarks to the Author):

The manuscript has been revised carefully, which could be accepted at its current form.

Reviewer #2 (Remarks to the Author):

The authors have addressed my concerns. I am ok with this revision.

Reviewer #3 (Remarks to the Author):

The authors have clearly worked hard to address the comments of all the reviewers and should be commended for their efforts. There are three outstanding issues that remain to be addressed.

We thank the Reviewer for recognizing our efforts to address all the comments of the Reviewers. Below, we provide the clarifications needed for the last three remaining points.

First, the authors do really need to address the recombination in the devices. If they are proposing the homojunction route as a way forward for efficient organic solar cells then it is not just the charge generation that is important but also the fate of the generated charges. Knowledge of this is critical.

We agree with the Reviewer that investigating the charge recombination is as important as investigating the charge generation. We are, however, surprised that the Reviewer feels that charge recombination has not been addressed thoroughly in our work.

Below we attempt a summary of what is included in both the Manuscript and Supplementary Information, including comments related to how this data reveals insights into the charge recombination in our devices:

1. **Electrical characterization (IV, EQE) of devices with mixed α -6T orientation (Figure 1):** EQE measurements give a first indication that charge generation efficiency is high in all the devices, independently of the selection of the buffer layer. Regarding recombination losses, the IV measurements show differences in the V_{OC} of the devices, indicating that further investigations are needed regarding the recombination mechanism in those devices.
2. **Subgap external quantum efficiency measurements (Supplementary Figure S2):** Charge generation and recombination are mediated by low-energy interfacial excited states, the charge-transfer (CT) states. We performed sensitive EQE measurements on the investigated devices in order to identify if the insertion of the buffer layer potentially leads to the formation of new CT-states of lower energy and, thus, the observed reduced V_{OC} 's in Figure 1. Figure S2 shows that the low energy edge (and hence the subgap energetics) of all devices is the same, indicating that the buffer layers do not form a low-energy interface that could drive recombination.

In order to emphasize better this finding in the text, we have now added the following sentence in page 4:

"...Except for the device with only BPhen, the sensitively measured EQE spectra of all other devices do not show any subgap absorption features, implying the absence of low-energy CT states originating from the α -6T/BL interface, which could drive charge generation and recombination (Supplementary Fig. S2)."

3. **Voltage loss analysis (Supplementary Table S2):** Knowing that the insertion of the buffer layers does not alter the subgap energetics of the devices, we then perform a voltage loss analysis in order to gain more insights into the different V_{OC} 's and quantify the radiative and non-radiative recombination. Since the gap in all devices is the same (Figure S2), they possess the same radiative voltage V_{rad} . This implies that the reduced V_{OC} 's are a result of increased non-radiative recombination ($\Delta V_{nonrad} = V_{rad} - V_{OC}$) occurring in the device.

This finding is highlighted in page 4 of the main text:

"...However, it should be noted that the use of extra BLs improves significantly the V_{OC} (up to 1.61 V) compared to the optical gap of α -6T (2.33 eV, Supplementary Fig. S1), mainly due to a reduction of the non-radiative recombination in those devices..."

4. **Exciton decay kinetics in α -6T devices and films (Supplementary Figures S7 and S8):** Transient absorption measurements on the α -6T based device (with rubrene as buffer layer) and the α -6T neat films revealed similar exciton decay

lifetimes, with the charge recombination being slightly faster in the device than the films.

This is highlighted in page 7 of the main text:

“Turning to the α -6T based HOSC with rubrene as BL, similar exciton decay lifetimes were observed (Fig. S7, S8). Except for a slightly faster charge recombination, all the spectral features are nearly identical with those in pristine α -6T thin films on ITO, indicating that efficient free charge carrier generation is not only present in devices but also in α -6T thin films. This confirms our conclusions above that device interfaces are not the origin of charge generation and indicates rather that this is a bulk phenomenon inherent to α -6T, one that is not usually observed in pristine organic materials.”

Additionally, we have performed analysis on the bimolecular recombination rate constant of charge carriers from the transient absorption decay for charges (at 780 nm). The bimolecular recombination rate constant was determined to be $1.9 \times 10^{-12} \text{cm}^3 \text{s}^{-1}$ when excited with $10 \mu\text{J cm}^{-2}$ and $3.3 \times 10^{-12} \text{cm}^3 \text{s}^{-1}$ when excited with $20 \mu\text{J cm}^{-2}$, which are relatively low in comparison with that observed in most bulk heterojunction organic solar cells ranging between 10^{-12} to $10^{-9} \text{cm}^3 \text{s}^{-1}$ (see references below) depending on the choice of materials and morphology. This suggests that using a single photoabsorber α -6T in homojunction solar cells does not accelerate the bimolecular recombination rate compared to the state-of-the-art non-fullerene based bulk heterojunction solar cells, as reported by others in the following references [Adv. Energy Mater. 2017, 7, 1701561 1701561; *Energy Environ. Sci.*, 2018,11, 3019-3032; *Energy Environ. Sci.*, 2020, Advance Article (<https://doi.org/10.1039/D0EE01338B>)].

Figure S6b. Analysis of bimolecular recombination rate constant from the transient absorption decay representing the bimolecular recombination of charges. For a bimolecular recombination following the rate equation of $\frac{d[n]}{dt} = -kn^2$, the rate constant (k) can be extracted from the slope by plotting $\frac{1}{[n]}$ against t (shown in open dots). In this case, the slopes were fitted to be $1.9 \times 10^{-12} \text{cm}^3 \text{s}^{-1}$ and $3.3 \times 10^{-12} \text{cm}^3 \text{s}^{-1}$ respectively under pump fluences of 10 and $20 \mu\text{J cm}^{-2}$.

Based on this, we have added this figure on page 6 of the Supplementary Info (Figure S6b) as well as the following sentence in the main text on page 6:

“The kinetics at 780 nm exhibit a strong dependence on the pump fluence, with decay dynamics accelerating at higher fluences characteristic of bimolecular recombination. Fitting the kinetics gives the bimolecular rate constant on the order of $10^{12} \text{ cm}^2 \text{ s}^{-1}$ (Fig. S6a), which is similar as that observed in bulkheterojunction OSCs.^{31,32} This fluence dependence therefore suggests that...”

Second, the authors should measure the charge mobilities, electron and hole of the films they are preparing - the assumption that they are essentially the same is bold.

Reviewer #2 has addressed the topic for charge mobilities of standing and lying phases in his/her Comment #6. We have provided values for hole mobilities in the 10^3 - $10^{-1} \text{ cm}^2/\text{Vs}$ range, reported for α -6T crystals and ordered films (Fichou, J. Mater. Chem. 10, 571, 2000; and references therein). Upon additional literature search, we have been able to find two recent papers [J. Appl. Phys. 103, 094509 (2008); Organic Electronics 13, 1614 (2012)] demonstrating good ambipolar transport properties in α -6T films, with mobilities in the 10^{-4} - $10^{-2} \text{ cm}^2/\text{Vs}$ range for both holes and electrons. These works demonstrated that α -6T, as many other molecular semiconductors, are intrinsically able to transport both holes and electrons.

The two above-mentioned references have been added to the main text, along with the following clarifying sentence:

“In a working device, hole and electron transport to electrodes would then take place in the standing and lying domains, respectively, exploiting the intrinsic ambipolar character of α -6T films.”

Moreover, we expressed our belief that the measurements of hole and electron mobilities in films of different or coexisting orientations are beyond the scope of this work: The key message of this work is that charge generation (and recombination) can happen in neat materials whose morphology consists of crystal domains with different orientations and energetics. In this concept, the use of a second material as electron acceptor becomes obsolete, thus, breaking the convention of bi-component organic solar cells functioning under the D-A concept. The already provided experimental data and state-of-the-art theoretical calculations draw strong conclusions that support our hypotheses. We are glad that Reviewer #2 seems to agree with this view.

Thus, electron and hole mobilities would be ‘nice-to-have’, complementary data to our work, but they are definitely not indispensable. Indeed, in the manuscript, we show data of neat 6T photovoltaic devices, which have reasonable FF and EQEs, showing that electron and hole mobilities are sufficiently high to obtain working devices. Charge carrier mobility measurements require quite a lot of time and resources but adding them would not alter any of the conclusions of the paper. Given the urgency of the topic, we therefore decided to omit them since they would significantly delay the publication of the paper.

Finally, it appears that the devices are all from a single batch with a chip containing four devices. If that is the case then there should be at least two batches produced to prove the statement that the process is reproducible.

The Reviewer is correct, the provided electrical data corresponded to devices existing on the same sample. With this data, we had the intention to point out the necessary reproducibility of our device fabrication method, the processing in vacuum with thermal evaporation.

In general, the high reproducibility of this method stems from the very controlled processing parameters (heating temperature, evaporation rate), together with the high purity of the resulting thin-films due to the processing in vacuum as well as the high purity of the evaporated materials. To ensure this high purity, every material batch has been sublimed twice in-house. Following the suggestion of the Reviewer, we provide here additional device data coming from different material batches of α -6T.

The new data are included in the Supplementary Table S7, and are highlighted in yellow. The devices shown in the main text, are those highlighted with bold letters.

Materials	Batch	Device	J_{sc} (mA cm ⁻²)	V_{oc} (V)	FF (%)	PCE (%)
α -6T/BPhen	A	1	1.33	1.25	41.4	0.7
		2	1.32	1.25	41.0	0.7
		3	1.33	1.25	41.3	0.7
		4	1.31	1.25	41.1	0.7
	B	1	1.28	1.25	44.1	0.7
		2	1.26	1.25	43.7	0.7
		3	1.26	1.25	43.8	0.7
		4	1.25	1.25	44.2	0.7
α -6T/TCTA	A	1	3.55	1.57	39.4	2.2
		2	3.56	1.56	39.7	2.2
		3	3.53	1.56	39.6	2.2
		4	3.49	1.57	39.1	2.1
	B	1	3.45	1.57	38.6	2.1
		2	3.50	1.56	38.1	2.1
		3	3.51	1.57	38.2	2.1
		4	3.48	1.57	38.4	2.1
α -6T/TBPe	A	1	3.00	1.41	42.5	1.8
		2	3.02	1.41	42.0	1.8
		3	3.03	1.41	42.1	1.8
		4	3.00	1.41	41.9	1.8
	B	1	2.98	1.41	41.9	1.8
		2	3.03	1.41	41.4	1.8
		3	3.05	1.41	41.6	1.8

		4	3.01	1.41	41.8	1.8
α -6T/C545T	A	1	3.82	1.46	34.7	1.9
		2	3.80	1.48	33.6	1.9
		3	3.79	1.46	33.5	1.8
		4	3.81	1.46	33.3	1.8
	B	1	3.64	1.46	33.1	1.8
		2	3.66	1.46	32.9	1.8
		3	3.70	1.47	32.4	1.8
		4	3.69	1.46	33.4	1.8
α -6T/DBzA	A	1	3.27	1.56	40.6	2.1
		2	3.33	1.57	39.9	2.1
		3	3.27	1.57	39.8	2.0
		4	3.33	1.56	39.4	2.0
	B	1	3.25	1.56	39.4	2.0
		2	3.32	1.57	39.1	2.0
		3	3.29	1.57	38.9	2.0
		4	3.27	1.57	38.5	2.0
α -6T/TPBI	A	1	3.45	1.50	33.3	1.6
		2	3.43	1.48	32.9	1.6
		3	3.44	1.50	33.0	1.6
		4	3.41	1.47	33.6	1.6
	B	1	3.33	1.50	32.7	1.6
		2	3.29	1.49	32.9	1.6
		3	3.25	1.48	33.4	1.5
		4	3.30	1.50	33.1	1.6
α -6T/TPBA	A	1	3.61	1.61	47.0	2.8
		2	3.58	1.60	45.5	2.7
		3	3.62	1.61	45.6	2.8
		4	3.63	1.58	46.3	2.7
	B	1	3.59	1.61	45.5	2.7
		2	3.55	1.60	45.2	2.7
		3	3.52	1.61	46.1	2.7
		4	3.50	1.61	45.9	2.7
α -6T/Rubrene	A	1	3.79	1.61	50.3	2.9
		2	3.80	1.61	50.2	2.9
		3	3.78	1.61	50.4	2.9
		4	3.75	1.61	50.1	2.8
	B	1	3.62	1.61	49.8	2.8
		2	3.65	1.61	49.5	2.8
		3	3.66	1.61	48.7	2.8
		4	3.63	1.61	48.6	2.8

Reviewers' Comments:

Reviewer #3:

Remarks to the Author:

I am content with the responses.